# Stringent test of QED with hydrogen-like tin

J. Morgner[1✉], B. Tu[1], C. M. König[1], T. Sailer[1], F. Heiße[1], H. Bekker[2], B. Sikora[1], C. Lyu[1], V. A. Yerokhin[1], Z. Harman[1], J. R. Crespo López-Urrutia[1], C. H. Keitel[1], S. Sturm[1] & K. Blaum[1]

Inner-shell electrons naturally sense the electric field close to the nucleus, which can reach extreme values beyond $10^{15}$ V cm$^{-1}$ for the innermost electrons[1]. Especially in few-electron, highly charged ions, the interaction with the electromagnetic fields can be accurately calculated within quantum electrodynamics (QED), rendering these ions good candidates to test the validity of QED in strong fields. Consequently, their Lamb shifts were intensively studied in the past several decades[2,3]. Another approach is the measurement of gyromagnetic factors ($g$ factors) in highly charged ions[4–7]. However, so far, either experimental accuracy or small field strength in low-$Z$ ions[5,6] limited the stringency of these QED tests. Here we report on our high-precision, high-field test of QED in hydrogen-like $^{118}$Sn$^{49+}$. The highly charged ions were produced with the Heidelberg electron beam ion trap (EBIT)[8] and injected into the ALPHATRAP Penning-trap setup[9], in which the bound-electron $g$ factor was measured with a precision of 0.5 parts per billion (ppb). For comparison, we present state-of-the-art theory calculations, which together test the underlying QED to about 0.012%, yielding a stringent test in the strong-field regime. With this measurement, we challenge the best tests by means of the Lamb shift and, with anticipated advances in the $g$-factor theory, surpass them by more than an order of magnitude.

In 1963, Richard Feynman called QED the greatest success in the physical sciences[10]. Describing the ubiquitous interactions of charges and the electromagnetic field with real and virtual photons, QED is the prime example of quantum field theories. Experimentally, QED has been tested with high stringency in low electromagnetic fields. Such tests are closely related to the determination of fundamental constants, such as, for example, the recent measurement of the $g − 2$ value, which allowed to extract the fine-structure constant $\alpha$ with a precision of $1.1 × 10^{-10}$ (ref. 11). By contrast, only a few experimental tests have been carried out at high electromagnetic field strengths. Here bound-state QED can yield high accuracy in the prediction of atomic and molecular systems. Thus, testing QED calculations still has wide implications for many branches of science.

In the past, muonic atoms have been studied extensively, leading to a series of stringent tests of the vacuum polarization in strong electric fields[12–14]. However, Standard Model predictions of muonic fine-structure splittings are inconsistent with experimental data[15–17]. Also, recently, the muon ($g − 2$) value has been remeasured and shows a 4.2$\sigma$ discrepancy[18]. As a consequence, this strongly motivates further tests of QED in strong electromagnetic fields.

Highly charged ions are an interesting candidate for such tests; because of the strong interaction between the (few) electrons and the nucleus, these systems also show enhanced sensitivity for potential new physics[19]. In these few-electron systems, the electric field experienced by the remaining electrons can exceed $10^{15}$ V cm$^{-1}$ (ref. 1), hence the electronic wavefunction is perturbed strongly, resulting in modified properties that can be measured and compared with theoretical predictions. So far, bound-state QED in high-$Z$ highly charged ions has been investigated most accurately by measurements of

the Lamb shift[20,21]. At present, calculations of the Lamb shift use an 'all-order' approach including all QED effects in one-loop and two-loop Feynman diagrams[22]. For testing bound-state QED using the magnetic moment or the $g$ factor of the bound electron, the theoretical approach is similar. Owing to the further interaction with a magnetic field, its calculation requires the inclusion of extra terms. But different to the Lamb shift, the calculation of the $g$-factor two-loop contributions with an all-order approach has not yet been completed. Therefore these contributions are calculated using a series expansion in $Z\alpha$, which is expected to have large uncertainty at high $Z$, owing to the strong scaling with $Z$. Here $Z$ is the atomic number and $\alpha$ is the fine-structure constant. In low-$Z$ systems, as the expansion coefficient $Z\alpha$ is small, high accuracy can be achieved in the prediction. Many systems with different charge states have been explored in the past[6,7,23–25]. Furthermore, the measurement of the hydrogen-like carbon $g$ factor allowed to determine the electron mass to an unprecedented precision[26]. The so far heaviest measured $g$ factor of hydrogen-like ions is $^{28}$Si$^{13+}$, which allowed for a stringent test of QED in low-to-medium-$Z$ ions[6,27].

Here we report on our high-precision $g$-factor measurement in hydrogen-like $^{118}$Sn$^{49+}$, reaching directly into the medium-to-high-$Z$ range. To achieve this, we produce the hydrogen-like ions externally in the Heidelberg EBIT[8], which can reach substantially higher charge states than the ion sources that were previously available for this type of measurement. From there, the ions are transported into the ALPHATRAP apparatus, in which we capture them to perform high-precision spectroscopy of the bound-electron $g$ factor. We further compare the measured value with its state-of-the-art theory prediction, which tests bound-state QED in a mean electric field of $1.6 × 10^{15}$ V cm$^{-1}$,

[1]Max-Planck-Institut für Kernphysik, Heidelberg, Germany. [2]Helmholtz-Institut Mainz, GSI Helmholtzzentrum für Schwerionenforschung, Mainz, Germany. ✉e-mail: Jonathan.Morgner@mpi-hd.mpg.de

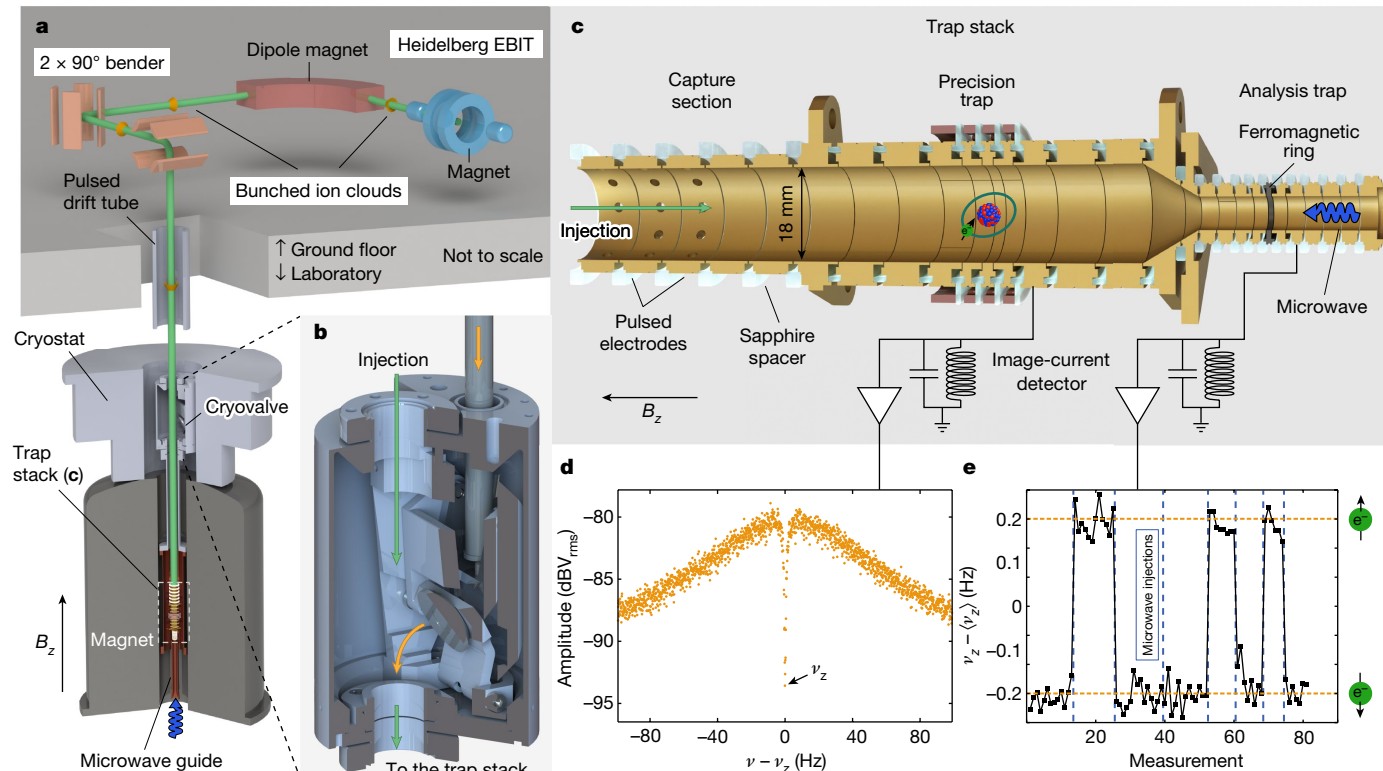

**Fig. 1 | Experimental setup for production, trapping and detection of hydrogen-like $^{118}\mathrm{Sn}^{49+}$. a**, The highly charged ions are produced in the Heidelberg EBIT. By means of a room-temperature beamline, the ions are transported into the ALPHATRAP magnet. **b**, The cryovalve allows to maintain an ultrahigh vacuum within the trap chamber. **c**, The 'trap stack' of the experimental setup. The ions are captured in the capture section by pulsing the applied voltage at the moment the ions are in the trap. Below is the precision trap, a seven-electrode trap in which the frequency ratio $\Gamma_0 = \nu_L/\nu_c$ is measured. An image-current detector is used to detect the particle motion in the trap. The voltage applied to the centre electrode is around −59 V. On the bottom of the trap stack, the analysis trap is located, which has a strong magnetic bottle, allowing the detection of the spin state of the bound electron. **d**, Fourier spectrum of the image-current detector with a $^{118}\mathrm{Sn}^{49+}$ particle in resonance. Fitting this 'dip' gives the axial frequency of the particle. **e**, Axial frequency change (about 300 mHz) after flipping the electron spin by microwave irradiation at the Larmor frequency.

60 times stronger compared with the $^{28}\mathrm{Si}^{13+}$ measurement, the so far strongest field for a precise *g*-factor measurement.

For the presented measurement, an enriched sample of Sn-118 was heated in an oven source for injection into the Heidelberg EBIT[8]. In the EBIT, a 200-mA electron beam focused by a 7-T magnetic field to a waist of a few tens of micrometres crosses the atomic beam in the centre electrode. With a kinetic energy of around 45 keV, well above the binding energy of the K-shell (≈35 keV (ref. 28)), electrons striking the tin atoms sequentially generate higher charge states until the charge-state distribution reaches a steady state. For the production and extraction of hydrogen-like $^{118}\mathrm{Sn}^{49+}$, a charge-breeding time of 60 s was used. After this, a fast pulse on the central electrode ejects the trapped highly charged ions. The ion bunch, with a kinetic energy of around 7 keV × $N_q$ ($N_q$ is the charge state), is transported through a room-temperature beamline, in which the required charge state is separated with a dipole magnet. A schematic view of the beamline is shown in Fig. 1a. Various ion-optical elements guide the ion cloud into the experimental setup. More details on the ion production can be found in Methods. Before entering the ALPHATRAP magnet, the ion bunch passes a pulsed drift tube, in which the kinetic energy is reduced to a few hundred eV × $N_q$, which is necessary to capture the ions in the trap. The cryogenic valve, shown in Fig. 1b, is opened briefly for the ion injection. This way, the inflow of gas from the room-temperature beamline is blocked, achieving an ultrahigh vacuum for long ion storage. For this measurement campaign, four hydrogen-like $^{118}\mathrm{Sn}^{49+}$ ions were loaded once. One of these was stored for three months, which allowed to precisely measure the magnetic moment of the bound electron.

The particles are trapped in our Penning-trap setup, which consists of a superconducting magnet with a *B* field of roughly 4 T for radial confinement. This is overlapped with an electrostatic field, which confines the ions in the axial direction. Once trapped, they are cooled by means of image currents to a temperature of 5.4(3) K. In the magnetic field, the Zeeman effect splits the energy levels of the electron spin. The energy difference is given as *h* times the Larmor frequency $\nu_L = (geB)/(4\pi m_e) \approx 107.6$ GHz, with *h* the Planck constant, *g* the *g* factor, *e* the electron charge and $m_e$ the electron mass. Furthermore, the free-space cyclotron frequency $\nu_c = (q_{ion}B)/(2\pi m_{ion}) \approx 25.7$ MHz governs the motion of the stored ion, in which $q_{ion}$ and $m_{ion}$ are its charge and mass, respectively. Because both result from the magnetic field *B*, their relation allows access to the *g* factor of the bound electron[29]:

$$g = 2\frac{\nu_L}{\nu_c}\frac{q_{ion}}{e}\frac{m_e}{m_{ion}} = 2\Gamma_0 N_q \frac{m_e}{m_{ion}}. \tag{1}$$

The charge ratio $N_q = q_{ion}/e$ is an integer number and the mass ratio is taken from other measurements[30,31]. This leaves the ratio $\Gamma_0 = \nu_L/\nu_c$, which has to be experimentally determined to extract the *g* factor. In the presented measurement, the double-trap method is used for its determination[32]. The 'trap stack' consists of two harmonic traps used for the measurement and an extra section for ion capture and storage (see Fig. 1c). In the precision trap, the three particle eigenmotions are determined. These are the modified cyclotron frequency $\nu_+ \approx 25$ MHz, the axial frequency $\nu_z \approx 650$ kHz and the magnetron frequency

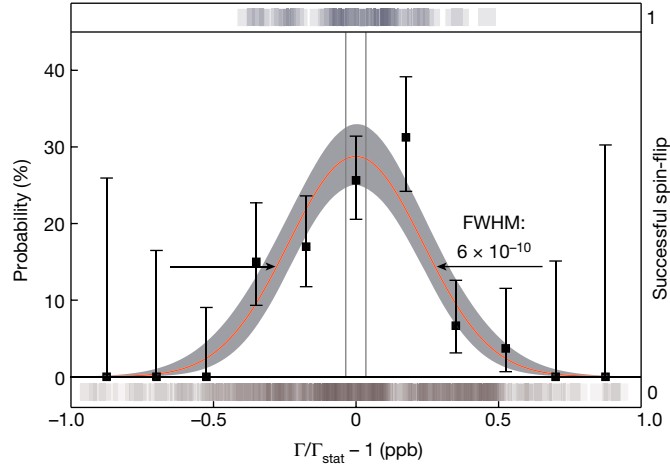

**Fig. 2 | Measured spin-flip resonance of the bound electron in $^{118}Sn^{49+}$.**
The maximum-likelihood fit is shown as the orange line, together with its corresponding $1\sigma$ error band (grey). The scattered points are used to guide the eye and represent a binned set of the data with 68% confidence levels given by a binomial fit. 68% confidence levels for the resonance centre are shown as vertical grey lines. The square shadows above represent the successful spin-flips in the precision trap, whereas the ones below show the unsuccessful attempts. FWHM, full width at half maximum.

$v_- \approx 8$ kHz. These are in direct relation to the free-space cyclotron frequency through the invariance theorem $v_c^2 = v_+^2 + v_z^2 + v_-^2$ (ref. 29).

The axial frequency is measured non-destructively by detection of image currents, induced by the moving particle next to the surrounding electrodes. If the particle is in thermal equilibrium, the noise spectrum of the cryogenic detector shows a distinct 'dip' (see Fig. 1d). The radial modes are detected by sideband coupling with the axial frequency, which enables the use of a single detector to measure all three frequencies. To measure the spin orientation, a second trap—called the analysis trap—is used. Its centre electrode is a ferromagnetic ring that produces a large quadratic coefficient of the magnetic field $B(z) = B_0 + B_1 z + B_2 z^2 + ...$, with $B_2 \approx 45$ kT m$^{-2}$. Described as the continuous Stern–Gerlach effect[33], the electron spin interacts with this so-called magnetic bottle, resulting in a spin-dependent axial force. Spin-flips caused by irradiating microwaves in resonance with the Larmor frequency $v_L$ can be detected in the analysis trap as a sudden change of the axial frequency, as shown in Fig. 1e. This magnetic-field inhomogeneity is problematic for a precise $\Gamma$ measurement only in this trap, hence two traps optimized for their respective use allows much higher precision.

A measurement cycle starts in the analysis trap by determining the spin state. Afterwards, the ion is adiabatically transported into the precision trap, in which the particle eigenmotions are measured. During the measurement of $v_+$, we irradiate a microwave at a random offset to the expected Larmor frequency. Then the ion is brought back into the analysis trap to examine whether the microwave injected in the precision trap has changed the spin orientation. By repeating this at different offsets around the expected $v_L$, we obtain a spin-flip probability as a function of the frequency ratio $v_L/v_c$. More details are given in Methods and the measurement scheme is shown in Extended Data Fig. 1. To determine the resonance parameters and their uncertainties, a maximum-likelihood analysis is performed. Several resonances have been recorded. Most of these were performed with different settings and are used to check systematic effects, such as the relativistic correction. For the extraction of $\Gamma_0$, only one is used, as it is the most precise with small motional radii and weak microwave power.

The scan consists of roughly 400 data points, of which 54 have been successful spin-flips. The binned data and the fit is shown in Fig. 2.

## Table 1 | Error budget

| Parameter | Relative shift (ppt) | Uncertainty (ppt) |
|---|---|---|
| $\Gamma_0 = v_L/v_c$ error budget: | | |
| $v_-$ measurement | – | 3.8 |
| Relativistic shift[54] | 23.7 | 4.8 |
| Image-charge shift[55] | 150 | 7.5 |
| $v_z$ line shape | – | 20 |
| Statistical uncertainty | – | 38 |
| $g$-factor error budget: | | |
| Total $\Gamma_0$ uncertainty | | 44 |
| Electron mass[26,31] | | 29 |
| $^{118}Sn^{49+}$ mass (this work) | | 475 |

The error budgets of $\Gamma_0$ and $g$ are shown. Further contributions are smaller than 1ppt, allowing to safely ignore them. More details can be found in the text and in Methods.

It is not saturated, that is, the maximum is well below 50%. Therefore, the resonance shape is mostly determined by magnetic-field jitter and not by power broadening. We use a Gaussian fit function to analyse the resonance. From this, $\Gamma_{stat}$ is extracted with a value of $\Gamma_{stat} = 4,189.05824237(16)$. The resulting ratio $v_L/v_c$ is corrected for systematic effects, arising from different sources as shown in the error budget in Table 1. Further details are explained in Methods. The corrected $\Gamma_0$ amounts to:

$$\Gamma_0 = 4,189.058241643(160)_{stat}(93)_{sys}. \qquad (2)$$

The parentheses represent the statistical and systematic uncertainty, respectively. Because the $g$ factor is also dependent on the mass of the highly charged ion, we also performed a cyclotron-frequency-ratio measurement to confirm the atomic mass evaluation (AME) value[30]. This yields a result of $m(^{118}Sn^{49+}) = 117.874869069(56)$ u, improving the value obtained from the AME (corrected for the missing electrons and their binding energies) by roughly a factor of ten.

We also calculate the electron-binding energies of neutral tin to extract the neutral-tin mass to similar accuracy. Details on the calculation and the mass measurement can be found in Methods. Using equation (1), we infer the $g$ factor to be:

$$g_{exp} = 1.910562058962(73)_{stat}(42)_{sys}(910)_{ext}. \qquad (3)$$

All uncertainties are $1\sigma$ confidence levels. The brackets are respectively the statistical and the systematic uncertainty, followed by the uncertainty of the external parameters, dominated by the atomic mass of tin-118. Although the $\Gamma_0$ uncertainty is $4.4 \times 10^{-11}$, the remaining mass uncertainty of the $^{118}Sn^{49+}$ ion limits the $g$ factor to a relative uncertainty of $4.8 \times 10^{-10}$.

The theoretical description of the free-electron $g$ factor is well established[31]. The dominant correction owing to the binding Coulomb potential of the nucleus is described by the Dirac value[34], $g_D - 2 = 4/3(\sqrt{1 - (Z\alpha)^2} - 1)$. Apart from that, binding corrections of QED Feynman diagrams with closed loops need to be taken into account. The non-relativistic QED approach, which treats the interaction between electron and nucleus perturbatively[35], cannot be expected to give good results for $Z = 50$ because the expansion parameter of this perturbation series, $Z\alpha$, is too large. Non-perturbative calculations for one-loop diagrams are well established[1,36], whereas the calculations for two-loop diagrams are only partially done[37–39].

The theory of the bound-electron $g$ factor has been previously tested in lighter ions, with $^{28}Si^{13+}$ being the heaviest hydrogen-like ion for which the $g$ factor has been measured[6,27]. In these previous measurements, one-loop binding corrections, namely the self-energy, the magnetic-loop vacuum polarization and the Uehling part of the

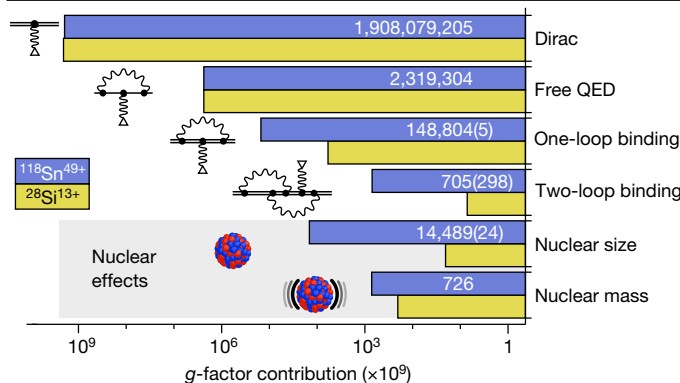

**Fig. 3 | g-factor contribution comparison of $^{28}$Si$^{13+}$ versus $^{118}$Sn$^{49+}$.** Because of the substantially higher $Z\alpha$, the binding corrections to the $g$ factor increase strongly from Si ($Z = 14$) to Sn ($Z = 50$). In the values without an uncertainty figure, all digits are significant.

electric-loop vacuum polarization corrections have been tested. In this measurement of $^{118}$Sn$^{49+}$, for the first time in a $g$-factor measurement, the Wichmann–Kroll part of the vacuum polarization correction is larger than the total theoretical and experimental uncertainties. Binding corrections to two-loop Feynman diagrams up to $\mathcal{O}((Z\alpha)^4)$ were tested in previous measurements of $^{28}$Si$^{13+}$ (refs. 6,27). Two-loop binding corrections of $\mathcal{O}((Z\alpha)^5)$, which were calculated only after the $^{28}$Si$^{13+}$ measurements[40,41], turn out to be smaller than our estimated uncertainty owing to uncalculated higher-order binding corrections. In Fig. 3, different theoretical contributions to the bound-electron $g$ factor of $^{118}$Sn$^{49+}$ are presented and compared with the $^{28}$Si$^{13+}$ $g$ factor. An extensive table, summarizing the different contributions, is given in the Extended Data Table 3.

Overall, we find a theoretical $^{118}$Sn$^{49+}$ $g$ factor of

$$g_{\text{theo}} = 1.910561821(299), \tag{4}$$

in agreement with the experimental value, although with a much larger uncertainty, which is dominated by uncalculated higher-order binding corrections $\mathcal{O}((Z\alpha)^6)$ to two-loop Feynman diagrams. Large-scale all-order calculations of these diagrams, which have the potential to greatly reduce the theoretical uncertainty, have been started in recent years[38,39].

Figure 4 shows the experimental against the theoretical uncertainty for different tests of bound-state QED in systems with high electromagnetic fields. So far, bound-state QED in heavy highly charged ions has been mostly tested by Lamb-shift measurements; the highest precisions were achieved in lithium-like systems[20,42–44]. With the tin measurement, the underlying bound-state QED is tested to about 0.20%. The total QED contribution, also including the zeroth order in the $Z\alpha$ expansion, is tested to about 0.012%. As the test is purely limited by the estimated uncertainty of the uncalculated higher-order two-loop terms, which is an order of magnitude larger than the uncertainty of effects from, for example, the finite nuclear size, the completion of continuing calculations can potentially improve the QED test markedly. Furthermore, an improved measurement of the atomic mass of the tin isotope could be achieved with higher precision by dedicated experiments (as shown in, for example, refs. 45,46), hence the experimental $g$-factor uncertainty can be reduced to that of $\Gamma_0$.

In conclusion, the $g$-factor measurement of hydrogen-like $^{118}$Sn$^{49+}$ paves the way for more sensitive tests of theoretical concepts and fundamental constants through Penning-trap-precision $g$-factor measurements of highly charged ions. It is a key step to the regime of strong fields previously uncharted for this kind of test. By combining the production capabilities of the Heidelberg EBIT

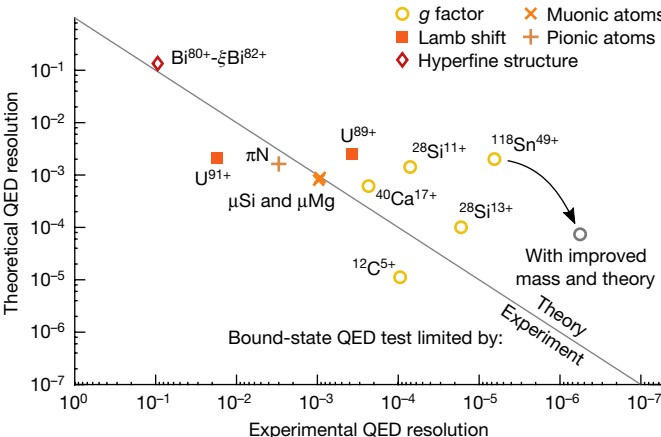

**Fig. 4 | Bound-state QED tests in high electric fields.** The uncertainty of experiment and theory relative to the bound-state QED contribution for certain highlighted measurements are shown[7,13,20,21,23,26,27,52,53]. The separate measurements are summarized in Extended Data Table 4. The muonic and pionic points are Lamb-shift measurements of the bound-muon/bound-pion energy levels in an excited transition. Values above the diagonal line would benefit from an improved theoretical calculation. In those below the line, the experimental error is the dominating uncertainty. The added limit with improved experiment and theory assumes a mass measurement that improves the $g$ factor to the $\Gamma_0$ uncertainty, as well as an improvement of the theoretical value to the limit imposed by the finite nuclear size uncertainty.

with the high-precision Penning-trap setup ALPHATRAP, we demonstrated the suitability for numerous future $g$-factor measurements with heavy highly charged ions[47,48]. Furthermore, it marks the first steps towards hyperfine spectroscopy in a heavy highly charged ion with unprecedented precision, which could be performed using a method similar to that demonstrated in the laser spectroscopy of $^{40}$Ar$^{13+}$ (ref. 49). Also, it is possible to measure different charge states and use a weighted-difference method to cancel finite size effects. This, together with an improved theory, would allow more stringent QED tests or possibly a determination of the fine-structure constant $\alpha$ (refs. 47,50,51).

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

## Methods

### Ion production

The ions produced in the EBIT are already rather slow (7 keV × $N_q$, with $N_q$ as the charge state), which simplifies the capture of ions after ejection. In a pulsed drift tube, the kinetic energy is reduced to roughly 500 eV × $N_q$, which is low enough that we can capture the ion bunch within the capture section of the trap. A detailed description about the ion capture can be found in ref. 9. For the injection from the Heidelberg EBIT, a lot of time was spent optimizing the production efficiency as well as the ion transport through the beamline. At present, the largest limit in the amount of ions that can be trapped in our experiment is the transport efficiency. From the EBIT, several tens of thousand hydrogen-like Sn-118 are ejected. In a single shot, because of poor transport efficiency in the beamline, at most a handful of particles can be trapped in our setup. The particles get lost during transport through the beamline. The phase-space distribution in the EBIT is rather large owing to the high temperature in the plasma. This results in a large spread of the ejected ions, which makes it difficult to efficiently transport and trap the whole ion bunch. Most ions are lost at the pulsed drift tube, in which only a fraction of the bunch is decelerated. As this is a technical limitation, improvement in the transport efficiency would also be feasible in the future. But because we perform the measurement mostly on single particles, we only have to trap a few particles, assuming that the vacuum of the trap is good. In total, the required setup time is about two weeks, with a following measurement time of several months. For future measurements in even heavier hydrogen-like systems, such as hydrogen-like lead, the electron-beam energy has to be increased further. As of now, the Heidelberg EBIT is limited to around 65 kV of acceleration voltage, making the ionization of higher binding energies impossible. When this is overcome, either by improving the Heidelberg EBIT or setting up an EBIT with enough electron-beam energy, the $g$-factor measurement is expected to be straightforward, as the measurement scheme for tin could also be used for hydrogen-like lead or uranium.

### Measurement scheme

After capture, the next step is to perform the high-precision measurement and corresponding systematic checks. As this normally takes a few weeks, the vacuum in our trap has to be extremely good. This is ensured by the cryogenic valve as shown in Fig. 1, which blocks the inflow of gas from the room-temperature beamline[9]. The valve and the cryogenic environment keep the vacuum below $10^{-16}$ mbar, allowing ion storage of many weeks.

In the presented $\Gamma_0$ measurement, a method similar to ref. 26 was used. The measurement sequence is shown in Extended Data Fig. 1. First, the initial spin state is investigated in the analysis trap. After adiabatic transport to the precision trap and a waiting time of two minutes to allow the voltages to settle, the modified cyclotron frequency is determined by sideband coupling. This is done by measuring and fitting of the noise spectrum that the detection circuit coupled to the ion produces[56]. Afterwards, the axial frequency is measured with the detection circuit as well; a spectrum is shown in Fig. 1d. This is followed by a 'pulse and amplify' (PnA) measurement sequence[57]. In PnA, we determine the accumulated phase of the modified cyclotron motion after a fixed evolution time. By initial excitation of the mode to a certain radius, a free evolution time for undisturbed phase accumulation followed by readout of the accumulated phase, we can track the phase of the particle over time, allowing to extract the frequency with high precision. In this measurement, the set of evolution times consists of five reference phase measurements with 0.2-s evolution time, two unwrapping phases and two 5.2-s evolution times, which are used for the determination of the magnetic field. In the reference phases, the magnetic-field jitter is negligible on the phase stability, as this is not yet dominating. For the long phases, the magnetic-field stability is the dominating jitter/drift source. The two unwrapping phases 0.5 s and 2.2 s are measured to allow a consistent phase unwrapping to the final 5.2-s measurement time. All except the two 5.2-s measurements are performed in a random order. The 5.2-s measurement is done twice at the end of the cycle. The first is used to precisely determine the magnetic field, which is used to calculate the expected Larmor frequency for the microwave injection. During the second 5.2-s measurement, a microwave with a random frequency offset to the expected $\Gamma$ is irradiated. With this, we measure the magnetic field one more time, while trying to perform a spin-flip with the microwave. For the final determination of the cyclotron frequency, we only use the five reference phases and the last 5.2-s phase during the microwave excitation. All others are used for unwrapping and to test certain systematics. After the PnA cycle, the axial frequency is measured a second time. The two axial-frequency measurements are used to identify systematic shifts in the measurement, although mostly the second is used for the calculation of the free-space cyclotron frequency, as it is immediately after the microwave injection. Afterwards, the ion is brought back to the analysis trap, in which we test if the precision trap microwave injection 'flipped' the spin. The measurement scheme is shown in Extended Data Fig. 1.

### Resonance analysis

The resulting resonance was analysed with a maximum-likelihood fit. Of the 387 spin-flip tries in the precision trap, 54 have been successful. The precision trap is at a position at which the second-order magnetic-field inhomogeneity is smaller than 10 mT m$^{-2}$; the influence on the field inhomogeneity can be safely neglected as the resulting error is less than 1 part per trillion (ppt)[58]. Odd-order inhomogeneity effects led by the first-order $B_1 \approx 2.64(3)$ mT m$^{-1}$ (ref. 9) are even further suppressed, as the odd orders are cancelled in a harmonic trap. The remaining influences in the line shape are given by magnetic-field jitter and by the power of the microwave injection.

Because the microwave injection time (5 s) is longer than the time the spin stays coherent to the (weak) drive, the spin-flip probability can be at most 50%. If high microwave powers are used, the chance to drive a spin-flip increases accordingly and broadens the resonance. A microwave-power-dominated resonance would follow a Lorentzian line shape. On the other hand, if the power is small enough, the magnetic-field jitter dominates, the spin-flip probability drops below 50% and the line shape changes from a Lorentzian to a Gaussian distribution. For the described resonance, a Gaussian function is used:

$$P(\Gamma) = A e^{-\frac{(\Gamma - \Gamma_{stat})^2}{2\sigma^2}}, \tag{5}$$

with $A$ being the amplitude, $\Gamma$ the irradiated Larmor frequency divided by the measured cyclotron frequency, $\Gamma_{stat}$ the centre of the resonance and $\sigma$ the standard deviation of the normally distributed data. Using a maximum-likelihood fitting method, which uses the unbinned dataset of $\Gamma$ ratios, we extract the parameters of the resonance. It has an amplitude of $A = 29(4)\%$ and a full width at half maximum of $5.6(3) \times 10^{-10}$. Owing to the low statistics, a Lorentzian fit results in a rather similar likelihood, but because the resonance width is consistent with the expected jitter based on the PnA phase stability, this is unlikely to be the case. Further, the Lorentzian fit would give a smaller uncertainty on the resonance centre. Using the Gaussian shape is thus the conservative approach in the extraction of $\Gamma_{stat}$ and its uncertainty. In Extended Data Fig. 2, the likelihood surfaces of the maximum-likelihood fit are shown. They show the maximum-likelihood planes in the three-dimensional parameter space spanned by the free parameters $\Gamma_{stat}$, $\sigma$ and $A$.

### Systematic effects

Of all the systematic effects, three dominate the uncertainty/correction. The largest shift is the image-charge shift. It is calculated according to ref. 55 and amounts to a relative shift of $1.50(8) \times 10^{-10}$. Here the error is 5% of the total value, following ref. 55.

The systematic effect with the largest uncertainty on $\Gamma_0$ arises from the axial frequency fit. The axial resonator causes a frequency pushing depending on the relative frequency difference of the particle and the resonator. This is accounted for in the fit function, but because the frequency of the resonator cannot be measured more accurately than a couple of hertz, the resulting uncertainty in the line shape of the axial dip cannot be neglected. With a resonator-frequency uncertainty of around 2 Hz, the fitted axial frequency changes by about 20 mHz. This translates to a $2.0 \times 10^{-11}$ relative uncertainty on $\Gamma_0$.

The relativistic correction owing to the mass increase on a cyclotron radius of 12.8(13) µm is $2.4(5) \times 10^{-11}$ (ref. 59). Here we assume an error of 10% on the cyclotron radius during PnA. This was also confirmed by measuring a resonance with 161 cycles, resulting in 18 spin-flips at a much higher cyclotron radius (102(10) µm). Owing to the stronger mass increase, this is shifted by roughly 1.6 ppb and is in agreement with the expected shift (see Extended Data Fig. 3). This resonance also tests for hypothetical systematic effects from the unwrapping, as—in this dataset—the longest PnA evolution time is set to 10.2 s, nearly twice as long as in the main resonance.

Owing to the exceptionally harmonic trapping field in our precision trap and a good magnetic homogeneity, the residual shift of the result from higher-order coefficients in the electromagnetic potential can be limited to less than 1 ppt.

Another uncertainty arises from the less frequent/less precise measurement of the magnetron frequency. This is only measured every several tens of cycles, as its influence on the free-space cyclotron frequency is small. Conservatively, we assume a maximum error of 0.3 Hz, which results in a relative uncertainty of $3.8 \times 10^{-12}$. An extensive table of the different systematic effects is given in the Extended Data Table 1.

## Mass measurement

To exclude the AME value of the neutral $^{118}$Sn isotope as an error source to the $g$ factor, we performed a direct mass measurement. We trapped a hydrogen-like $^{118}$Sn$^{49+}$ ion along with a hydrogen-like $^{12}$C$^{5+}$ ion. Both have a relatively similar $q/m$ value, which results in suppressed systematic effects in the measurement. The axial frequencies are separated by roughly 820 Hz for the same electric potential in the precision trap. To determine the mass, we measure the cyclotron frequency ratio (CFR), which allows to extract the mass ratio without precise knowledge of the magnetic field:

$$\frac{\nu_c(^{12}C^{5+})}{\nu_c(^{118}Sn^{49+})} = \frac{q(^{12}C^{5+})}{q(^{118}Sn^{49+})} \frac{m(^{118}Sn^{49+})}{m(^{12}C^{5+})} = \frac{5}{49} \frac{m(^{118}Sn^{49+})}{m(^{12}C^{5+})}. \quad (6)$$

To determine the mass, we used two measurement methods with partially different systematic effects to cross-check the result. A sketch of the measurement schemes in shown in Extended Data Fig. 4. In both methods, we measure the ions in an interleaved manner, transporting each particle subsequently into the precision trap, while storing the other in a neighbouring section. In the main method, we use the PnA scheme to determine the modified cyclotron frequency without any line-shape uncertainty. Atypically, we perform this measurement with one particle on resonance and the other 820 Hz apart. That way, we can apply the same voltages to the electrodes and effects resulting from a shifted potential/ion position are avoided. For this, we put the carbon ion in resonance; as a result of lower charge, this couples less to the detector system. It would be virtually impossible to detect its image currents when detuned by that amount. But because of the stronger coupling in hydrogen-like tin, this is still detectable even 820 Hz detuned, allowing to measure the motional frequencies.

In the second method, we change the electrostatic potential to tune each axial frequency into resonance with the detector. We use sideband coupling, also known as a 'double-dip' measurement, to measure the modified cyclotron frequencies of each particle. To move each particle on resonance, the axial potential is changed by roughly 150 mV out of

the initial $V \approx -59$ V. As possible patch potentials on the electrodes are not necessarily symmetric, the trap centre might shift slightly owing to the different voltages. Combined with a magnetic-field gradient $B_1 = 2.64(3)$ mT m$^{-1}$, the measured frequency ratio could be systematically shifted because of this possible misalignment.

Another possible systematic shift arises from the different line shapes of the ions on the detector. The dip width of hydrogen-like tin and hydrogen-like carbon differs by about a factor of ten. Thus, effects owing to an erroneous input parameter to the fitting model of the 'double dip' could result in a systematic effect, as each ion would be affected differently. In the PnA method, the modified cyclotron frequency is determined independent of the detector line shape, therefore removing the connected systematic uncertainties. Owing to the better understanding of the systematic effects in the PnA method, we use the 'double-dip' measurement solely as a cross-check of the measured mass. When considering all systematic effects and uncertainties, this agrees with the PnA measurement.

Note that, here, only five frequency ratios have been collected in the phase-sensitive measurement. Therefore, we conservatively use the standard deviation as our statistical uncertainty and not the standard deviation of the mean.

Correcting the measurement for relativistic effects and the image-charge shift, we get a final value for the mass ratio:

$$\frac{m(^{118}Sn^{49+})}{m(^{12}C^{5+})} = 9.8251510645(39)_{stat}(27)_{sys}. \quad (7)$$

The two brackets represent the statistical and systematic uncertainties, respectively. The mass of the $^{12}$C$^{5+}$ ion can be expressed in relation to the neutral carbon atom as unit of mass after accounting for the missing electrons and binding energies. This gives a mass value for the $^{12}$C$^{5+}$ ion of 11.99725768029217(43)(8) u (refs. 28,31). The brackets are the uncertainty of the binding energies and the electron-mass uncertainty, respectively. As the total uncertainty is less than 0.01 ppt, its influence on the resulting hydrogen-like tin mass can be safely neglected. From this, we can infer a value for the atomic mass of the hydrogen-like tin-118 ion:

$$m(^{118}Sn^{49+}) = 117.874869069(47)_{stat}(32)_{sys} \text{ u}. \quad (8)$$

The first bracket is the statistical uncertainty and the second bracket is the systematic uncertainty, which is dominated by the relativistic effect.

## Binding energies and neutral mass

The AME2020 value for the mass of neutral $^{118}$Sn is 117.90160663(54) u, which has an uncertainty of 466 eV/$c^2$ ($u = 9.3149410242(28) \times 10^8$ eV/$c^2$). With the measured mass of $^{118}$Sn$^{49+}$, we can improve the accuracy of the mass of the neutral atom through

$$m(^{118}Sn) = m(^{118}Sn^{49+}) + 49m_e - \Delta E/c^2. \quad (9)$$

Here $m_e = 0.000548579909070(16)$ u is the electron rest mass[31]. $\Delta E$ is the energy required to ionize the 49 electrons from a neutral Sn atom and is theoretically calculated to be 132,746(5) eV. The final result is $m(^{118}Sn) = 117.901606974(56)(5)$ u, which is a factor of 9.5 more accurate than the previous best value.

The value $\Delta E$ is derived from the electron-binding-energy difference between neutral Sn and hydrogen-like Sn$^{49+}$. As the electron-binding energy of Sn$^{49+}$ is known to be 35,192.501(11) eV (ref. 22), we only need to calculate the electron-binding energy of neutral tin. This is performed with an ab initio, fully relativistic, multiconfiguration Dirac–Hartree–Fock (MCDHF) together with a relativistic configuration interaction (RCI) method[60–62] implemented in the GRASP2018 code[62]. However, because the binding energy of the four outermost electrons of Sn has

been experimentally determined to be 93.22(4) eV (ref. 28), the binding energy of the ground state of Pd-like $Sn^{4+}$ ($[Kr]4d^{10}\,^1S_0$) is calculated instead. We note that the ionization potential for the fifth electron is also known experimentally. However, the ground state of $Sn^{5+}$ is an open-shell configuration with a total angular momentum of 5/2. Thus, it requires a much larger basis set and amount of computational power to achieve the same accuracy as for the closed-shell ion $Sn^{4+}$.

Within the MCDHF scheme, the many-electron atomic-state function is constructed as a linear combination of configuration state functions (CSFs) with common total angular momentum ($J$), magnetic ($M$) and parity ($P$) quantum numbers: $|\Gamma PJM\rangle = \sum_k c_k |\gamma_k PJM\rangle$. The CSFs $|\gamma_k PJM\rangle$ are given as $jj$-coupled Slater determinants of one-electron orbitals and $\gamma_k$ summarizes all the parameters needed to fully define the CSF, that is, the orbital occupation and coupling of single-electron angular momenta. $\Gamma$ collectively denotes all the $\gamma_k$ included in the representation of the atomic-state function. The mixing coefficients $c_k$ and the radial orbital wavefunctions are obtained by solving the MCDHF equations self-consistently[60,61], including the Dirac–Coulomb Hamiltonian. After that, the RCI method is used to calculate the contributions from mass shift, transverse photon interactions and QED effects.

We start with a Dirac–Hartree–Fock (DHF) calculation, in which only the ground-state configuration is considered. This gives a binding energy of 167,973.14 eV, with a −4.42-eV correction from the finite nuclear size effect. Further calculation with RCI adds contributions of −0.57 eV from mass shift, −120.34 eV from frequency-independent transverse photon interactions (or Breit interactions), 1.16 eV from frequency-dependent transverse photon interactions and −79.08 eV from QED terms. To derive the electron correlation energy, the size of the CSF basis set is gradually expanded through single and double (SD) excitation of electrons from the ground-state configuration to high-lying virtual orbitals. This allows us to monitor the convergence of the correlation energy by adding and optimizing virtual orbitals layer by layer up to $n = 10$ ($n$ is the principal quantum number), with all orbital angular momenta ranging from 0 to $n − 1$ being included.

Six terms may lead the error bar: the uncertainty in the nuclear parameters, the finite basis set, the uncounted higher-order electron correlations, the insufficient basis functions, the inaccurate estimations for QED corrections and the uncalculated QED effect to the mass shift. The uncertainty in the nuclear radius gives a 0.06-eV uncertainty to the corrections in the finite nuclear size effect. To exclude the error caused by the finite basis set, we extrapolate our calculated value to $n = \infty$. This results in a SD correlation energy of 67.26(23) eV for the ground state of $Sn^{4+}$. As this analysis is performed under the RCI calculation, the contribution from Breit interactions is fully accounted for. The frequency-dependent transverse photon interaction cannot be included in the multiconfiguration calculations of GRASP2018, thus its uncertainty will be accounted for later together with other untreated minor effects when deriving the systematic errors. The relativistic mass-shift operator implemented in the GRASP2018 code is accurate to the order of $(m_e/M)(\alpha Z)^4 m_e c^2$ ($M$ is the mass of the nucleus). Therefore, it bears an uncertainty of 0.02 eV for the mass-shift correction.

In the GRASP2018 code, the QED corrections are estimated by means of a screened-hydrogenic approximation[62]. This correction is dominated by inner-shell electrons. With an absolute value of 79.08 eV for $Sn^{4+}$, one already has a QED correction of 76.64 eV for Be-like $Sn^{46+}$. Fortunately, the QED effect for a nearby element, Be-like $Xe^{50+}$, has been known to a sub-eV accuracy through ab initio QED calculations[63,64]. This allows us to infer the accuracy of our QED calculations: with a calculated value of 99.03 eV for $Xe^{50+}$, it is 0.85 eV larger than its ab initio result. Assuming the same relative deviation, we derive a QED correction of 78.40(68) eV for $Sn^{4+}$. However, for such an ion, the many-electron QED effects are difficult to evaluate accurately. In the following, we will effectively include these contributions into the systematic errors.

The systematic errors caused by the uncalculated effects can be estimated from the ionization potentials of the outermost electrons

of Sn. For example, the ionization potential of the 5s electron in $Sn^{3+}$ ($[Kr]4d^{10}5s\,^2S_{1/2}$) is determined to be 40.74(4) eV by experiment[28]. With the calculated binding energy of $Sn^{3+}$ under a similar SD excitation scheme, we derive an ionization potential of 40.07 eV, which is 0.67 eV smaller than the experimental value. This deviation mainly originated from the high-order correlation effects, many-electron QED effects, insufficient basis functions and frequency-dependent transverse photon interactions. Nevertheless, this deviation becomes smaller for highly charged ions[45]. Therefore, one could conservatively assume that the corrections to the ionization potentials decrease linearly to 0.10 eV for Cu-like $Sn^{21+}$ (because we have found that the deviation is already below 0.10 eV for the ionization potential of Cu-like $Kr^{6+}$). For the ionization potentials of ions throughout $Sn^{21+}$ to $Sn^{48+}$, we conservatively assume that they all have a correction of 0.10 eV. In total, the contribution from all unaccounted for terms is in the range 0–9.73 eV. To cover this whole range, we can add a correction of 4.86 eV to the total binding energy of $Sn^{4+}$ and simultaneously assume a systematic error of 4.86 eV.

Finally, we arrive at a total binding energy of 167,847(5) eV for the ground state of $Sn^{4+}$. The different contributions and their uncertainties are summarized in Extended Data Table 2. This gives a binding-energy difference of 132,748(5) eV between neutral Sn and hydrogen-like $Sn^{49+}$. Combining the measured mass for $Sn^{49+}$, we obtain

$$m(^{118}Sn) = 117.901606974(56)_{\mathrm{exp}}(5)_{\mathrm{theo}}\ \mathrm{u} \tag{10}$$

for the neutral tin-118 atom. The first bracket is the measurement uncertainty and the second is the uncertainty of the electron-binding energies.

## Theory of the bound-electron g factor

The leading $g$-factor contribution was first calculated in ref. 34. It is based on the approximation of an infinitely small and infinitely heavy nucleus. Therefore, corrections owing to the finite nuclear size and mass need to be taken into account. Furthermore, QED corrections contribute to the total $g$ factor, just as in the case of the free electron. However, QED corrections in the case of the bound electron differ from the free-electron case. In Fig. 4, we highlight especially QED binding corrections, that is, the difference between QED corrections for bound and free electrons. In the following, we discuss all relevant contributions.

**Free-electron contributions.** Contributions to the free-electron $g$ factor were taken from ref. 31, namely, the one-loop to five-loop QED contributions. Hadronic as well as electroweak contributions as given in ref. 31 are too small to be relevant in this work.

**Nuclear corrections.** The finite size (FS) correction to the $g$ factor as given in Extended Data Table 3 was calculated for the two-parameter Fermi distribution using formulas and tabulated parameters from ref. 65. The first uncertainty corresponds to the nuclear root mean square charge radius as given in ref. 66. Note that the uncertainty corresponding to the number of digits specified for relevant parameters in ref. 65 is much smaller than the radius uncertainty, $7 \times 10^{-10} \ll 2 \times 10^{-8}$. The second uncertainty of the FS correction from the table is a combination of the uncertainty owing to the nuclear polarization, deformation and susceptibility, together with a conservative estimate of the nuclear model dependence. The model dependence is the leading uncertainty and expresses the difference of the FS corrections for the two-parameter Fermi and the homogeneously charged sphere model, again following ref. 65. A direct calculation of the FS correction for the sphere model using semianalytic wavefunctions was consistent with the result from ref. 65.

Calculating the FS correction analytically using formulas from refs. 67,68, we find a disagreement with numerical results corresponding to three times the nuclear radius uncertainty for both the sphere and Fermi models. This suggests that further, higher-order contributions

need to be determined for the analytic approach to be accurate at high $Z$, such as $Z = 50$.

Finally, we also calculated the FS correction using the GRASP code[62]. In the absence of results for $^{118}\mathrm{Sn}^{49+}$ in refs. 69,70, the nuclear polarization correction was estimated as zero following ref. 65, with an uncertainty estimated as 50% of the nuclear model uncertainty of the FS correction. Our estimates for the extra uncertainty owing to nuclear deformation[71] and nuclear susceptibility[72] corrections are negligible.

For recoil calculations, we used the mass value presented in this paper. The leading recoil term of the first order in the mass ratio was calculated to all orders in $Z\alpha$ using formulas and tabulated parameters in ref. 73. Recoil corrections of higher order in $\frac{m}{M}$ were calculated to the leading order in $Z\alpha$ but exactly in the mass ratio[74–76]. We find the result up to $\mathcal{O}\left(\left(\frac{m}{M}\right)^2\right)$ (refs. 77,78) to deviate by less than $1 \times 10^{-14}$ from the all-order result.

Radiative recoil corrections were calculated using formulas from refs. 1,74,76–78. So far, recoil corrections to the $g$ factor have been derived only for the model of a point-like nucleus.

**One-loop QED.** Binding corrections at the one-loop level have been calculated to all orders in $Z\alpha$. For the one-loop self-energy correction, we used the result from ref. 36, which is based on the model of a point-like nucleus. We calculated the Uehling part of the electric-loop vacuum polarization diagram for the model of a point-like nucleus both numerically, using the Uehling potential from ref. 79 and the bound-electron wavefunction perturbed by a constant external magnetic field as derived in ref. 80, as well as analytically using formulas from ref. 81, with both results in excellent agreement. Values for the Wichmann–Kroll electric-loop vacuum polarization correction as well as the magnetic-loop vacuum polarization correction were taken from ref. 1 and were calculated for extended nuclei in that work.

**Combined QED-FS corrections.** Bound-electron QED corrections, when carried out for the model of an extended nucleus, give slightly different results compared with point-nucleus calculations. Therefore, for an accurate theoretical description of the bound-electron $g$ factor, we have to take into account the correction to QED contributions owing to the finite nuclear size (QED-FS corrections). As mentioned in the previous section, the Wichmann–Kroll part of the electric-loop vacuum polarization as well as the magnetic-loop vacuum polarization corrections already include the QED-FS corrections. We calculated the FS corrections to the one-loop self-energy correction and the Uehling contribution to the electric-loop vacuum polarization correction for the two-parameter Fermi distribution of the nuclear charge using formulas, and tabulated parameters for $Z = 50$, as given in ref. 65. An older calculation of the FS correction to the one-loop self-energy contribution based on the spherical shell model of the nucleus from ref. 36 differs by $2.1 \times 10^{-9}$ from the result for the Fermi distribution. Using semianalytic wavefunctions for the homogeneous sphere model of the nucleus, we calculated the FS correction to the Uehling part of the electric-loop vacuum polarization correction[82]. Our result differs by $5 \times 10^{-10}$ from the result for the Fermi distribution. We therefore assign (nuclear model) uncertainties of $2.1 \times 10^{-9}$ to the one-loop self-energy FS correction, as well as $5 \times 10^{-10}$ to the FS correction to the Uehling part of the electric-loop vacuum polarization correction.

**Two-loop and higher QED diagrams.** We calculated binding corrections to two-to-five-loop QED diagrams of order $(Z\alpha)^2$ following ref. 83. See also refs. 74,77,78 for earlier derivations of these binding corrections. (Results given in the lines '2-loop QED, $(Z\alpha)^2$' and '≥3-loop QED, binding' in Extended Data Table 3.)

Two-loop binding corrections of order $(Z\alpha)^4$ were derived in refs. 40,84. All-order calculations in $Z\alpha$ were carried out in ref. 37 for a subset of two-loop diagrams, namely those diagrams with at least one vacuum polarization loop. However, magnetic-loop vacuum polarization diagrams were not considered in that work. In ref. 37, results are given explicitly for the contribution of orders $(Z\alpha)^5$ and higher. We give this result in the line '$(Z\alpha)^{5+}$ S(VP)E, SEVP, VPVP'. For the remaining two-loop Feynman diagrams, QED corrections of order $(Z\alpha)^5$ were calculated in ref. 40. In ref. 40, a relative uncertainty of 13% is mentioned, owing to uncalculated Feynman diagrams contributing to order $(Z\alpha)^5$. This corresponds to our uncertainty in the line $(Z\alpha)^5$ in Extended Data Table 3. The uncertainty in the line '$(Z\alpha)^{5+}$ S(VP)E, SEVP, VPVP' corresponds to higher-order corrections of order $(Z\alpha)^{6+}$ of two-loop Feynman diagrams with one vacuum polarization magnetic loop, by interpolating between tabulated results from ref. 39 for nearby $Z$.

The uncertainties owing to uncalculated Feynman diagrams with two self-energy loops of order $(Z\alpha)^{6+}$ were estimated using the methods from refs. 40,84, with the larger of the two estimates chosen as the uncertainty for these contributions in Extended Data Table 3. Furthermore, uncalculated binding corrections of $\mathcal{O}((Z\alpha)^{4+})$ to Feynman diagrams with three and more loops were estimated by adapting the method in ref. 84 (uncertainty in the line '≥3-loop QED, binding').

As can be seen from Extended Data Table 3, the total uncertainty of the theoretical $g$-factor value is dominated by uncalculated higher order in $Z\alpha$ two-loop QED contributions. Calculations to improve the accuracy of two-loop corrections are underway[38,39]. To improve the theoretical accuracy, an all-order (in $Z\alpha$) calculation of the two-loop QED correction is required. Such calculations are underway[38,39]. The most difficult part of the calculation is the two-loop self-energy, which is split into several parts according to the degree of their ultraviolet divergence, which are known as the loop-after-loop (LAL) correction and the F, M and P terms[85].

The F term is the part with overlapping ultraviolet divergences. It can be represented by Feynman diagrams with only free-electron propagators inside the self-energy loops and is evaluated in momentum space, thus avoiding any partial-wave expansion.

The M term is the ultraviolet finite part of the two-loop self-energy correction with Coulomb Dirac propagators inside the self-energy loops. Because the Coulomb Dirac propagator is best known in coordinate space, M-term calculations need to be carried out using a coordinate representation. Typically, M-term contributions are a double infinite sum of partial waves over angular momentum quantum numbers, which requires a very large number of partial waves to be calculated in practice. The calculation of every partial wave requires a multidimensional integration to be carried out numerically.

The P term is the part of the two-loop self-energy correction that contains Coulomb Dirac propagators in one part of the Feynman diagrams, as well as an ultraviolet divergent subdiagram. This requires P-term contributions to be calculated in a mixed coordinate–momentum representation, which involves the numerical Fourier transforms of the Coulomb Dirac propagators over one of the radial arguments. Details can be found in our earlier work[38].

Results for the so-called LAL and F-term contributions to two-loop self-energy corrections have been obtained, with the uncertainty of the LAL correction given in ref. 38 for $Z = 50$ being $6.5 \times 10^{-9}$ and the uncertainty of the F-term being orders of magnitude better. The calculation of the remaining parts of the two-loop self-energy correction, that is, the M and P terms, is continuing. A notable improvement of the total theoretical uncertainty compared with the value given in Extended Data Table 3 can be anticipated once these calculations are complete.

**Muonic and hadronic vacuum polarization.** The result for the muonic vacuum polarization correction given in Extended Data Table 3 corresponds to the Uehling part of the electric-loop contribution, calculated for the model of a point-like nucleus. Comparing results for point-like nuclei with results for extended nuclei from ref. 86, we find good agreement between both for low $Z$. For $Z = 70$, the result for the extended nucleus is only about 50% of the point-nucleus result. For even higher $Z$, the extended nucleus result is much smaller than

50% of the point-nucleus result. For $Z = 50$, we therefore expect the muonic vacuum polarization correction for an extended nucleus to be larger than 50% of the point-nucleus result and assign a 50% uncertainty to the muonic vacuum polarization correction. We also calculated the Uehling part of the muonic vacuum polarization correction for the sphere model of the nucleus[82]. Our result of $-2.0 \times 10^{-9}$ is, within the specified uncertainty, in agreement with the point-nucleus result from Extended Data Table 3.

The hadronic vacuum polarization correction ('Hadronic Uehling') was estimated following refs. 86,87 as 0.671 and 0.664 times the muonic vacuum polarization correction, with both estimates being identical to all digits given. We assigned the same uncertainty to the hadronic vacuum polarization correction that was used for the muonic vacuum polarization correction.

## Data availability

The datasets generated and/or analysed during this study are available from the corresponding author on reasonable request.

## Code availability

All code used for the analysis and production of results of this study are available from the corresponding author on reasonable request.

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

**Acknowledgements** We acknowledge help from M. Rosner and N. Rehbehn with the operation of the Heidelberg EBIT. We also thank N. Oreshkina for discussion of the paper and its content. This work was supported by the Max Planck Society (MPG), the International Max Planck Research School for Quantum Dynamics in Physics, Chemistry and Biology (IMPRS-QD), the German Research Foundation (DFG) Collaborative Research Centre SFB 1225 (ISOQUANT) and the MPG-PTB-RIKEN Center for Time, Constants and Fundamental Symmetries. This project has received funding from the European Research Council (ERC) under the European Union's Horizon 2020 research and innovation programme under grant agreement number 832848 FunI. This work comprises parts of the PhD thesis work of C.M.K. and J.M. to be submitted to Heidelberg University, Germany.

**Author contributions** J.M. and B.T. contributed equally to the ion production with the help of H.B. and J.R.C.L.-U. The experiment was maintained and performed by J.M., C.M.K., T.S., B.T., F.H. and S.S. The data were analysed by J.M., F.H. and B.T. Theoretical calculations were performed by B.S., C.L., Z.H. and V.A.Y. J.M. and B.S. contributed to the writing of the manuscript, which was edited by S.S., Z.H., J.R.C.L.-U. and K.B. All authors discussed and approved the data as well as the manuscript.

**Funding** Open access funding provided by Max Planck Society.

**Competing interests** The authors declare no competing interests.

**Additional information**
**Correspondence and requests for materials** should be addressed to J. Morgner.

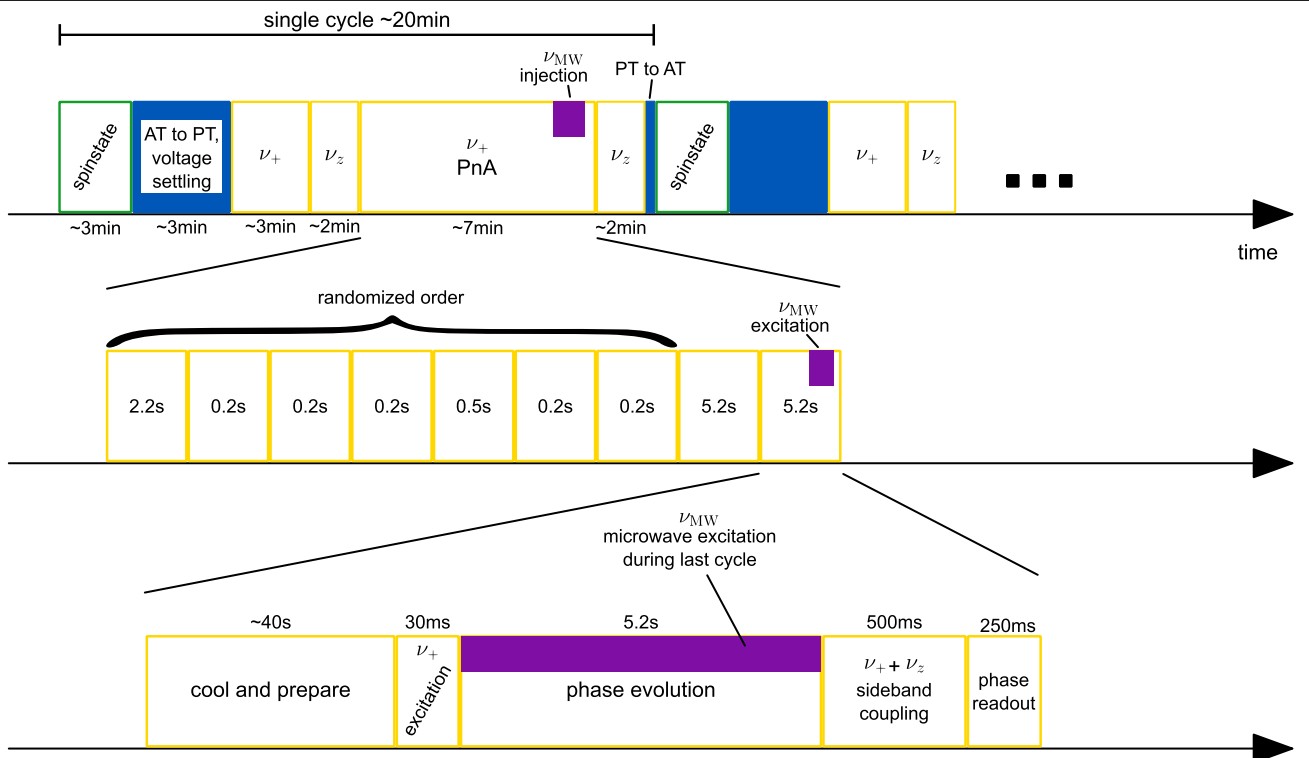

**Extended Data Fig. 1 | Γ measurement scheme.** The ratio $\Gamma = \nu_{\mathrm{t}}/\nu_{\mathrm{c}}$ is measured with the scheme shown. $\nu_+$ is measured most precisely through the PnA method. $\nu_-$ is measured only every couple of cycles, as only moderate precision is required for a precise $\nu_{\mathrm{c}}$ measurement. In the PnA method, different evolution times are used. The first few are randomized, whereas the last two are always with 5.2-s evolution times. The first precisely determines the current $\nu_{\mathrm{c}}$, which is used to guess the $\nu_{\mathrm{c}}$ for the microwave injection with higher accuracy.

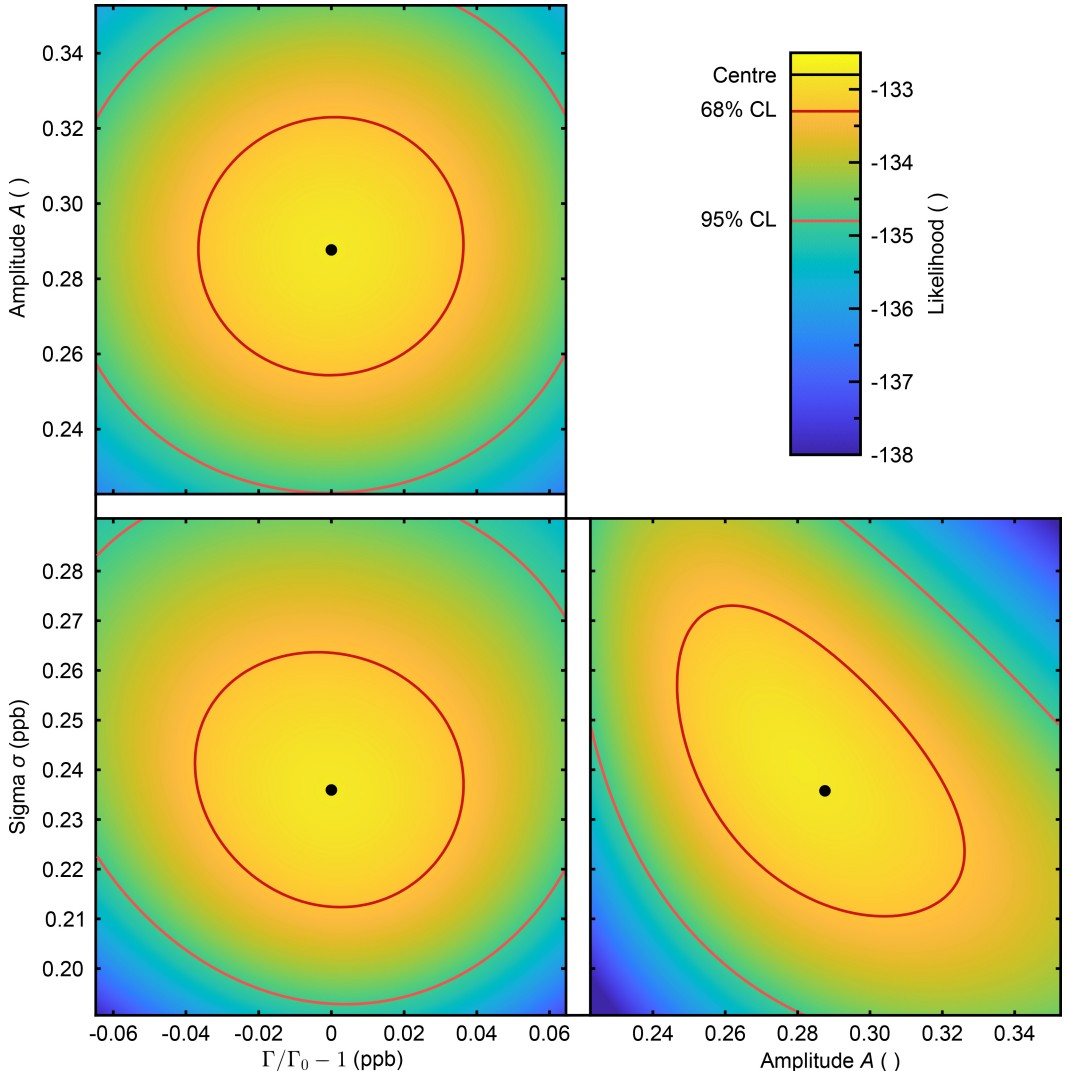

**Extended Data Fig. 2 | Likelihood surfaces.** The likelihood planes through the optimal parameter set (black circles) are shown. In the $\Gamma_0$ dimension, the surfaces show a circular shape, indicating little to no correlation with the other parameters. Overall, the fit converged correctly to the global maximum-likelihood position.

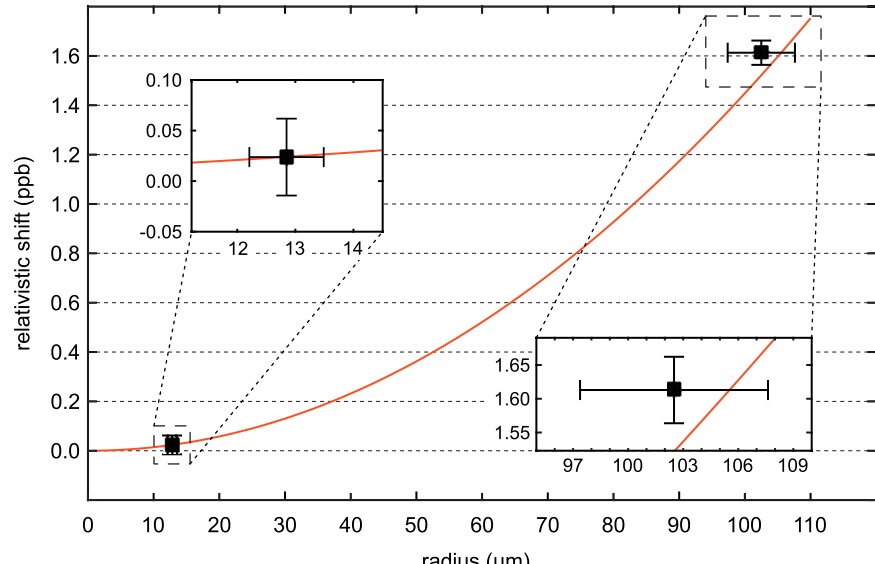

**Extended Data Fig. 3 | Relativistic shift.** Two resonances with different cyclotron radii have been measured. One with a cyclotron radius of 12.8(13) µm and the other with a cyclotron radius of 102(10) µm. Both resonances are in agreement with the total relativistic shift for a given radius (red line).

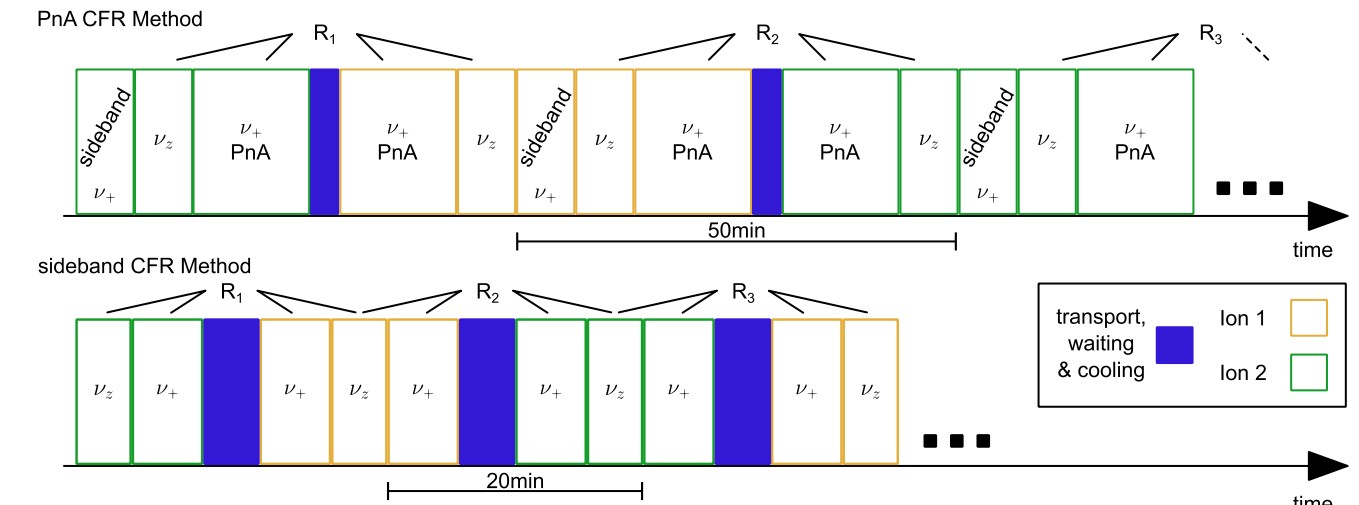

**Extended Data Fig. 4 | CFR measurement scheme.** Two different measurement schemes have been used for the mass determination. In the PnA scheme, each ratio has their own axial frequency measurement. The $\nu_+$ sideband detection is used in the PnA cycles to avoid wrong phase unwrapping. Apart from this, it is not used in the CFR evaluation. The sideband CFR method consists of interleaved 'double-dip' measurements, with axial dips in between. As the ions are switched every cycle, each axial dip measurement is used for both neighbouring $\nu_+$ measurements.

**Extended Data Table 1 | Experimental results**

| Fit parameter | value |
|---|---|
| Amplitude | 29(4)% |
| Full-Width–Half-Maximum | $5.6(3) \times 10^{-10}$ |
| $\Gamma_{\text{stat}}$ | 4,189.05824235(16) |
| **Systematic shifts** | **relative shift** |
| Image charge shift | $1.50(8) \times 10^{-10}$ |
| Relativistic shift | $2.37(48) \times 10^{-11}$ |
| **Systematic uncertainties** (not shifts) | **relative uncertainty** |
| $^{118}$Sn mass (this work) | $4.8 \times 10^{-10}$ |
| Electron mass [31] | $2.9 \times 10^{-11}$ |
| Axial dip line shape | $2.0 \times 10^{-11}$ |
| $\nu_-$ frequency uncertainty | $3.8 \times 10^{-12}$ |
| Electrostatic anharmonicity $C_4, C_6, \ldots$ | $< 6 \times 10^{-14}$ |
| $B_2$ line shape | $< 6 \times 10^{-14}$ |

The fit parameters for the resonance are given, as well as the dominating systematic effects and the dominating uncertainties for the resulting g factor.

**Extended Data Table 2 | Binding energy**

| | $DHF_0$ | FNS | Breit | $\omega$TP | MS | QED | SDc | HO | total |
|---|---|---|---|---|---|---|---|---|---|
| $E_{Sn^{4+}}$ | 167,977.60 | -4.42 | -120.34 | 1.16 | -0.57 | -78.40 | 67.26 | 4.86 | 167,847.15 |
| $\delta E_{Sn^{4+}}$ | NaN | 0.06 | NaN | NaN | 0.02 | 0.68 | 0.23 | 4.86 | 4.92 |

Different contributions to the binding energy of $Sn^{4+}$: $DHF_0$, the DHF energy assuming a point-like nuclear charge; FNS, finite nuclear size effect; Breit, the frequency-independent transverse photon interaction; $\omega$TP, the frequency-dependent transverse photon interaction calculated with DHF wavefunctions; MS, the mass shift; QED, the QED estimation based on screened-hydrogenic approximation; SDc, the correlations energy arising from single and double electron exchanges; HO, the systematic effect summarizing all other uncounted terms. The values of the $DHF_0$, Breit and $\omega$TP terms are dependent on basis. Such a basis dependency is lifted after taking into account all correlation effects. Thus, their uncertainties are effectively accounted for in the uncertainties of the SDc and HO terms. All values are shown in units of eV with two decimal digits. However, in the main text, the total binding energy is rounded up to an integer value, that is, 167,847(5) eV.

**Extended Data Table 3 | Theoretical *g* factor**

| Contribution | $^{118}\text{Sn}^{49+}$ | Reference |
|---|---|---|
| Dirac value | 1.908 079 205 3(1) | [1, 34] |
| Finite size | 0.000 014 489 4(110)(211) | TW, methods [62, 65-72] |
| 1-loop QED $(Z\alpha)^0$ | 0.002 322 819 5 | [31] |
| SE binding | 0.000 182 170 1 | [36] |
| SE-FS | -0.000 000 159 4(21) | [1, 36, 65] |
| VP-EL,Uehling | -0.000 035 203 4 | [1, 79, 81, 82] |
| VP-EL,Uehling FS | 0.000 000 113 7(5) | [1, 65, 82] |
| VP-EL, WK | 0.000 000 658 4 | [1] |
| VP-ML | 0.000 001 225 0(40) | [1, 89] |
| muonic VP | -0.000 000 002 7(14) | TW, methods [82, 86] |
| 2-loop QED $(Z\alpha)^0$ | -0.000 003 544 6 | [31, 90, 91] |
| $(Z\alpha)^2$ | -0.000 000 078 6 | [74, 77, 78, 83] |
| $(Z\alpha)^4$ | -0.000 001 123 8 | [84, 92] |
| $(Z\alpha)^5$ | 0.000 000 211 0(215) | [40, 41] |
| $(Z\alpha)^{5+}$ S(VP)E, SEVP, VPVP | 0.000 000 286 5(53) | [37, 39] |
| $(Z\alpha)^{6+}$ SESE | 0.000 000 000 0(2968) | [40, 84] |
| $\geq$ 3-loop QED $(Z\alpha)^0$ | 0.000 000 029 5 | [31, 93-97] |
| binding | 0.000 000 000 7(105) | [74, 77, 78, 83, 84] |
| Recoil $\frac{m}{M}$, all-order $Z\alpha$ | 0.000 000 726 8 | [1, 28, 62, 73] |
| h.o. $\frac{m}{M}$ | -0.000 000 000 1 | [74, 76-78] |
| rad-rec | -0.000 000 000 5(4) | [1, 74, 76-78] |
| Hadronic Uehling | -0.000 000 001 8(14) | TW, methods [86, 87, 98, 99] |
| Sum | 1.910 561 821 0(2988) | |
| Experiment | 1.910 562 059 0(9) | |

Contributions to the bound-electron $g_e$ factor in $^{118}\text{Sn}^{49+}$. (See refs. 88–99). 'TW' refers to results calculated in this work.

**Extended Data Table 4 | Bound-state QED tests**

| Observable | System | BS-QED contrb. (theo) | Experimental accuracy |
|---|---|---|---|
| Electronic Lamb shift | | | |
| | $U^{91+}$ [21] | 265.07(53) eV [22] | 4.6 eV |
| | $U^{89+}$ [20] | 41.77(10) eV [100, 101] | 0.015 eV |
| Muonic Lamb shift | | | |
| | $\mu$Mg [13] | 178.68(16) eV | 0.17 eV |
| | $\mu$Si [13] | 274.90(22) eV | 0.27 eV |
| Pionic Lamb shift | | | |
| | $\pi$N [53] | 1.364 5(19) eV * | 0.0050 eV |
| Hyperfine splitting | | | |
| | spec.diff.[52] | 0.229(30) meV [102] | 0.022 meV |
| g factor | | | |
| | $^{12}C^{5+}$ [26] | $0.843\ 376\ 2(98) \times 10^{-6\ \dagger,\ddagger}$ | $7.3 \times 10^{-11}$ |
| | $^{28}Si^{13+}$ [27] | $5.855\ 954(589) \times 10^{-6\ \ddagger}$ | $9.9 \times 10^{-11}$ |
| | $^{28}Si^{11+}$ [23] | $0.981\ 34(28) \times 10^{-6}$ [103] | $1.4 \times 10^{-10}$ |
| | $^{40}Ca^{17+}$ [7] | $2.355\ 74(29) \times 10^{-6}$ [103] | $1.1 \times 10^{-10}$ |
| | $^{118}Sn^{49+}$ | $0.148\ 802(300)(11) \times 10^{-3}$ | $9.1 \times 10^{-10}$ (mass unc.) |
| | | | $8.4 \times 10^{-11}$ ($\Gamma_0$ unc.) |

Values for the data points in Fig. 4. (See refs. 100–103). *In this measurement, the pion mass is extracted from the experiment–theory comparison. To account for that in the QED test, the theoretical uncertainty is adjusted to the previous literature value of the charged pion mass. †Similarly, this measurement is used to determine the electron mass such that a test of QED can only be as accurate as the accuracy of an independent electron-mass measurement. Recently, the electron mass has been confirmed with similar accuracy in HD+ spectroscopy[104], allowing to (basically) use the full experimental precision for a test of QED. ‡Calculated similar to the $^{118}Sn^{49+}$ theory g factor as described in Methods.