## [Peer Review File · Nature]

Manuscript Title: Stringent test of QED with hydrogenlike tin

Reviewer Comments & Author Rebuttals

Reviewer Reports on the Initial Version:

Referees' comments:

Referee #1 (Remarks to the Author):

The efforts to verify QED have never stopped. In this paper, the authors present a high-precision, high-field test of QED in hydrogenlike $118\text{Sn}49+$ by using the ALPHATRAP Penning-trap setup. In electronic systems, bound-state QED in high-Z atoms has been probed most accurately by measurements of the Lamb shift before. Here, the authors do the test by measuring the electron g-factor in $118\text{Sn}49+$. The electric field is 60 times stronger compared to that in the last $28\text{Si}13+$ measurement, the so far strongest field for a precise g-factor measurement. It is very exciting that this measurement shows the highest experimental QED resolution, and what's more exciting is the authors can measure more g-factors with heavier elements in the next. I believe this work is a very important step, leads to more breakthrough results soon. Besides, this paper is well written. Thus, I recommend it for the publication in Nature journal.

I have some optional suggestions or questions.

1. Figure 4, would it be possible that the authors provide a detailed table in e.g., Extended data part or so, showing each data point's specific raw data, both theoretical and experimental ones, etc. Since it may take some time for the readers to look and seek the raw data one by one from Ref. [13, 19, 21, 22, 39-41], and calculated them one by one after then.
2. I am very much looking forward to seeing the measurements of heavier elements, e.g., Pb or Bi? Would there be any extra difficulties?
3. Extended Data Table 3, I am not well-versed in the calculations of QED, now the difference between the experiment and the calculation of g-factor is $2380\text{E}-10$, smaller than the uncertainty of the calculation $2988\text{E}-10$. Line 769, what if the difference would be larger than the significant improved uncertainty once these calculations are complete?
4. Line 413, "In a single shot...at most a handful of particles can be trapped in our setup." What limits the transport efficiency? How many particles are best needed in a single shot?

Referee #2 (Remarks to the Author):

In the present manuscript, the authors present a record-breaking measurement of bound electron g factor in a medium/heavy hydrogen-like ion. The present measurement of the bound electron g-factor results in one of the most stringent tests for the most studied fundamental interaction, quantum electrodynamics, in the presence of the extremely strong Coulomb field generated by the atom nucleus. The previous record measurement of bound electron g-factor has been performed in 2011 in hydrogen-like silicon (with nuclear charge $Z=14$). The present article reports a measurement in hydrogen-like tin, which has a significantly higher nuclear charge ($Z=50$) that allow for a much more stringent test of the state-of-the art predictions of bound-system quantum electrodynamics. Elusive quantum electrodynamics effects are in facts scaling as Z to the power of 4. A g-factor measurement with such a heavy hydrogen-like ion has been made possible by the coupling of two cutting-edge devices: the Heidelberg EBIT ion source and the ALPHATRAP Penning trap. Previously, highly charged ions were created directly in the trap, but with limiting ionizing power methods. The coupling of this Penning trap with such an ion source is just unique. The adopted methodology is well established and mastered by the authors and it is the best that can be found in the field. The statistics treatment of the data analysis is rigorous and well

presented. I particularly appreciated the choice of the conservatory approach for evaluating certain systematic errors, which tendentially enlarge the final error bars, but that provides a high reliability of the final result.

The manuscript is very well written with a clear structure, with a very attractive abstract that well anticipate the results presented in the body text, with a pleasant introduction with a large panorama on the most recent measurements in the field and with an impactful conclusion that well resume the key features and implications of the presented measurement. The more technical details are provided in the "methods" additional sections, clearly presented also for a reader not familiar with the field. The bibliography references are very complete and adapted, including the most recent articles of the literature.

Because of the unicity of such a measurement, its record accuracy and its impact on the field and its implication in our general knowledge on fundamental interactions, the present article deserves to be published in Nature.

Before that, I suggest however the following improvements.

- Data analysis and fig.2: Because of the very small number of failing or successful measurement of the spin-flip, I guess that the error bars of the "to guide the eye" scattering points presented in fig. 2 are based on Poisson distribution. This should be clearly indicated. More generally, independently on the chosen profile (Gaussian or Lorentzian), more details should be provided on the likelihood function itself with respect to the number of failing and successful spin-flip attempts for each analyzed frequency.

- Line 275, table 3 and theoretical methods: If all-orders one-loop calculations are well established (line 275), why such results are presented as $(Z\alpha)$ expansion in table 3? This is confusing and deserves more explanations. Can the authors in addition indicate briefly the main difficulties of the determination of all-order two-loops terms? Last but not the least, what is meaning of F-, M- and P-terms?

A list of minor suggestions is following.

- Line 57: Reference [11] should be updated. The cited preprint article is now published.

- Line 72: reference [56] should be cited here in support

- Line 86: Which theoretical approaches are considered for the comparison? All-orders or with an expansion in $(Z\alpha)$? It is not clear and, even if this is discussed in more details later, it should be shortly indicated here in the introduction.

- Some technical aspects are missing in the methods. What is the typical vacuum?

Moreover, a reference about the cryogenic valve should be added. Without this component, the experiment would be impossible.

Referee #3 (Remarks to the Author):

First of all, the g-factor measurement presented in the paper constitutes a milestone in this field of research. It crowns many years of effort by the MPIK experimental group. Indeed it challenges the bound-state QED theory as strongly as ever. The following remarks concern the presentation of the results. Taking into account the broad audience of Nature, it is especially important to give a comprehensive view of research in this field.

1. Motivation of investigations in this field is not limited by the tests of QED. I suggest at least to stress the relation to such topics as search for new physics and determination of fundamental constants (including their possible variation) and nuclear properties. For the sake of a broader picture, it is also important to mention previous g-factor measurements for highly charged ions (there are not so many, each one is important). Possibly, high-precision measurements in more complicated systems, e.g. neutral cesium and singly charged calcium, could be mentioned.

2. The measurement procedure largely resembles the previous Penning-trap g-factor

measurements, in particular, those accomplished at ALPHATRAP setup. This is fine, but I would be mostly interested by the special features of this one. What was the largest problem when going from low to high Z? Which special techniques allowed the authors to solve it successfully? How far are we from measurements with e.g. lead and uranium?

3. In Figure 4 the level of testing bound-state QED is demonstrated for binding energies (H- and Li-like uranium), HFS (H- and Li-like bismuth) and g-factor - only with H-like systems for some reason. And what about Li-like in this case? Moreover, nothing is said about muonic and pionic atoms - only a couple of dots on the plot. What is the measured quantity in this case?

4. In section 2.6 the calculations of the electron binding energy in Sn within the MCDHF and RCI methods are discussed. First, I suggest giving a hint about how the QED effects are taken into account in this case, i.e., what is the model operator implemented in GRASP. Second, I'm skeptical about the corresponding uncertainty estimation via the ab initio results, since the many-electron effects may be dominant in the case of Sn^{4+} . Luckily, this is not the dominant contribution to the total uncertainty. Third, is the omega-dependence of the Breit interaction taken into account? And finally, a table with all the discussed contributions and uncertainties would be quite useful here.

5. The g-factor theory part 2.7 is rather chaotic and difficult to comprehend by a non-specialist. It is split into three paragraphs "Nuclear corrections", "Uncertainties of two- and higher-loop QED", and "Muonic and hadronic vacuum polarization". I would expect a step-by-step discussion of the QED effects, from the leading terms to the unknown ones. Instead, only some problematic parts are discussed in an unpredictable order. In general, I suggest to reshape this part, in particular, to comment somehow on all the contributions in Table 3. In the following, some more specific comments are given.

6. The first and largest part "Nuclear corrections" is expected to cover the nuclear size (NS), nuclear recoil (NR) and nuclear polarization (NP) effects. The NS discussion is the most elaborate, and still I can't understand, whether it includes the NS-QED and NS-NR cross terms? Meanwhile, NR is incomplete: nothing is said about the first-order term in the mass ratio. Some aspects of the vacuum polarization are discussed here, while it's not a "nuclear correction", or is it?

7. What was the motivation to pay some special attention to the muonic and hadronic vacuum polarization? Is it strongly enhanced in this case for some reason? Will it be possible to verify these corrections in future improved measurements?

8. Table 3 contains lines with references and those with "adopted from". What is the difference? Which terms were calculated in this work?

9. I'm not aware of the corresponding Nature style requirements, but a separate part without any text looks strange to me. Section 2.8 with several tables would benefit strongly from some comments concerning the tables. Otherwise these tables may belong to the corresponding section in which they are referenced.

Author Rebuttals to Initial Comments:

Response Letter

Answers to the Referees are marked in **blue**

Text in the Manuscript that has been modified/added is highlighted in red.

Referee #1 (Remarks to the Author):

The efforts to verify QED have never stopped. In this paper, the authors present a high-precision, high-field test of QED in hydrogenlike $118\text{Sn}49+$ by using the ALPHATRAP Penning-trap setup. In electronic systems, bound-state QED in high-Z atoms has been probed most accurately by measurements of the Lamb shift before. Here, the authors do the test by measuring the electron g-factor in $118\text{Sn}49+$. The electric field is 60 times stronger compared to that in the last $28\text{Si}13+$ measurement, the so far strongest field for a precise g-factor measurement. It is very exciting that this measurement shows the highest experimental QED resolution, and what's more exciting is the authors can measure more g-factors with heavier elements in the next. I believe this work is a very important step, leads to more breakthrough results soon. Besides, this paper is well written. Thus, I recommend it for the publication in Nature journal.

I have some optional suggestions or questions.

1. Figure 4, would it be possible that the authors provide a detailed table in e.g., Extended data part or so, showing each data point's specific raw data, both theoretical and experimental ones, etc. Since it may take some time for the readers to look and seek the raw data one by one from Ref. [13, 19, 21, 22, 39-41], and calculated them one by one after then.

Great idea, we added a table to the Extended Data.

2. I am very much looking forward to seeing the measurements of heavier elements, e.g., Pb or Bi? Would there be any extra difficulties?

The difficulty lies mostly in the production and injection of the highly charged ions.

These are limited to the acceleration voltage of the electron beam that can be applied in the Heidelberg EBIT.

A few sentences are added to the Methods section "Ion Production" to explain this in the text. (445 f.)

In contrast, the cryogenic vacuum and the spin-flip detection fidelity seem to be sufficient for even the heaviest species in our current setup. For the future we are currently working on the Hyper-EBIT with which we aim towards much higher beam energies for (near) arbitrary charge state production.

3. Extended Data Table 3, I am not well-versed in the calculations of QED, now the difference between the experiment and the calculation of g-factor is $2380\text{E}-10$, smaller than the uncertainty of the calculation $2988\text{E}-10$. Line 769, what if the difference would be larger than the significant improved uncertainty once these calculations are complete?

The Referee describes a very interesting scenario. If such a deviation is discovered after the completion of two-loop QED calculations, two scenarios can occur. For once there could be an error in either the experiment or in the theoretical calculations. The second scenario would be that the unexplained discrepancy is due to physics beyond the standard model.

To resolve such a tension, further measurements of highly charged ion g factors should be performed to confirm/disprove the discrepancy in a separate system, where effects would possibly scale differently.

Such a discrepancy exists in example for the g factor of the free muon, where there is a long-standing discrepancy between theory and experiment on the level of 3-5 sigma. This deviation is widely considered as one of the promising possible signs of new physics beyond the Standard Model.

4. Line 413, "In a single shot...at most a handful of particles can be trapped in our setup." What limits the transport efficiency? How many particles are best needed in a single shot?

We added some sentences that explain this in the "Ion Production" section of the Methods. (435f)

Referee #2 (Remarks to the Author):

In the present manuscript, the authors present a record-breaking measurement of bound electron g factor in a medium/heavy hydrogen-like ion. The present measurement of the bound electron g -factor results in one of the most stringent tests for the most studied fundamental interaction, quantum electrodynamics, in the presence of the extremely strong Coulomb field generated by the atom nucleus. The previous record measurement of bound electron g -factor has been performed in 2011 in hydrogen-like silicon (with nuclear charge $Z=14$). The present article reports a measurement in hydrogen-like tin, which has a significantly higher nuclear charge ($Z=50$) that allow for a much more stringent test of the state-of-the art predictions of bound-system quantum electrodynamics. Elusive quantum electrodynamics effects are in facts scaling as Z to the power of 4. A g -factor measurement with such a heavy hydrogen-like ion has been made possible by the coupling of two cutting-edge devices: the Heidelberg EBIT ion source and the ALPHATRAP Penning trap. Previously, highly charged ions were created directly in the trap, but with limiting ionizing power methods. The coupling of this Penning trap with such an ion source is just unique.

The adopted methodology is well established and mastered by the authors and it is the best that can be found in the field. The statistics treatment of the data analysis is rigorous and well presented. I particularly appreciated the choice of the conservatory approach for evaluating certain systematic errors, which tendentially enlarge the final error bars, but that provides a high reliability of the final result.

The manuscript is very well written with a clear structure, with a very attractive abstract that well anticipate the results presented in the body text, with a pleasant introduction with a large panorama on the most recent measurements in the field and with an impactful conclusion that well resume the key features and implications of the presented measurement. The more technical details are provided in the "methods" additional sections, clearly presented also for a reader not familiar with the field. The bibliography references are very complete and adapted, including the most recent articles of the literature.

Because of the unicity of such a measurement, its record accuracy and its impact on the field and its implication in our general knowledge on fundamental interactions, the present article deserves to be published in Nature.

Before that, I suggest however the following improvements.

- Data analysis and fig.2: Because of the very small number of failing or successful measurement of the spin-flip, I guess that the error bars of the “to guide the eye” scattering points presented in fig. 2 are based on Poisson distribution. This should be clearly indicated. More generally, independently on the chosen profile (Gaussian or Lorentzian), more details should be provided on the likelihood function itself with respect to the number of failing and successful spin-flip attempts for each analyzed frequency.

The scattered points and their error bars are calculated using a binomial distribution of the amount of successful and unsuccessful spin flips in the corresponding bin/range.

Ultimately, the likelihood fit uses each data-point as a separate value. No binning takes place in order to fit the measured datapoints. Additionally, we assume each individual point to have no "external" (other than Bernoulli/random) error, so each is weighted the same. The statistical measurement uncertainty of the individual points is thus encoded in the resonance width.

Furthermore, to allow the interested reader to follow the evaluation more closely, the likelihood surfaces for the three free parameters are added to the extended data in Fig. 6.

A few explanatory lines have been added to the “Resonance analysis” section (524f and 532f)

- Line 275, table 3 and theoretical methods: If all-orders one-loop calculations are well established (line 275), why such results are presented as $(Z\alpha)$ expansion in table 3? This is confusing and deserves more explanations. Can the authors in addition indicate briefly the main difficulties of the determination of all-order two-loops terms? Last but not the least, what is meaning of F-, M- and P-terms?

We presented individual $Z\alpha$ expansion contributions for the one-loop QED correction in order to facilitate the analysis for the two-loop QED correction, which is calculated only within the $Z\alpha$ expansion so far. However, we do see the Referee’s point that such a split is confusing (and indeed unnecessary), especially in the high-Z regime. We therefore rearranged the one-loop contributions in Table 3 into “free QED” and “all-order binding corrections”. The latter are split according to different effects, namely “self-energy”, “vacuum polarization - electric loop - Uehling”, “vacuum polarization - electric loop - Wichmann Kroll” and “vacuum polarization - magnetic loop”. For a transparent comparison between both versions of Table 3: The $O((Z\alpha))^2$ contribution is a pure “self-energy” effect, and the $O((Z\alpha))^4$ contribution is the sum of a “self-energy” part of $-0.000\ 126\ 259\ 5$ and a “vacuum polarization - electric loop - Uehling” part of $-0.000\ 043\ 912\ 2$.

Brief explanations of the F-, M- and P-terms have been added to the text, along with a very brief description of the main difficulties in the calculation of each of them.

A list of minor suggestions is following.

- Line 57: Reference [11] should be updated. The cited preprint article is now published.

done

- Line 72: reference [56] should be cited here in support

done

- Line 86: Which theoretical approaches are considered for the comparison? All-orders or with an expansion in $(Z\alpha)$? It is not clear and, even if this is discussed in more details later, it should be shortly indicated here in the introduction.

The complete bound-state QED contribution is included, independent of the means of calculation.

We thank the referee for pointing out that this was indeed not presented well in the introduction; therefore, we modified some parts there to clarify that the calculation of the bound-state QED part of the g factor includes both all-order calculations (for the one-loop), and $Z\alpha$ expansion for the two loop contributions.

The changes include the added sentence in line 80f. and the removal of the sentence in the introduction talking about the ongoing calculations of the all order approach.

By this we want to emphasize that the presented calculation of the g factor employs a $Z\alpha$ expansion for the two-loop contribution.

- Some technical aspects are missing in the methods. What is the typical vacuum?

Moreover, a reference about the cryogenic valve should be added. Without this component, the experiment would be impossible.

Great point, to emphasize this more, we added some sentences to the start of the Methods section "Measurement scheme"

Referee #3 (Remarks to the Author):

First of all, the g -factor measurement presented in the paper constitutes a milestone in this field of research. It crowns many years of effort by the MPIK experimental group. Indeed it challenges the bound-state QED theory as strongly as ever. The following remarks concern the presentation of the results. Taking into account the broad audience of *Nature*, it is especially important to give a comprehensive view of research in this field.

1. Motivation of investigations in this field is not limited by the tests of QED. I suggest at least to stress the relation to such topics as search for new physics and determination of fundamental constants (including their possible variation) and nuclear properties. For the sake of a broader picture, it is also important to mention previous g -factor measurements for highly charged ions (there are not so many, each one is important). Possibly, high-precision measurements in more complicated systems, e.g. neutral cesium and singly charged calcium, could be mentioned.

We thank the referee for the valuable suggestions to improve the introduction part. In the version of the manuscript at hand, we had tried to implement those relations in the motivation and we have talked about the $g-2$ measurement, although we did not mention the connected α determination.

To accommodate the referee's suggestions, we modified the introduction slightly to put more emphasis on the closely related determination of fundamental constants.

To bring the aspect of New Physics into the introduction, we added one more sentence in line 67f. This also serves to introduce highly charged ions in the Main text as well, thus far, it was only brought up in the abstract.

To highlight the previously measured highly charged ion g factors better in the text, the sentence in line 86f. has been added, and the following sentence has been modified to suit the change.

Discussing the measurements on many-electron systems such as Cs and Ca⁺ as suggested by the referee would be surely interesting, but adding it would require an additional section within the introduction, as this is already quite long, and due to strict space constraints for the *Nature Magazine*, we decided against it.

2. The measurement procedure largely resembles the previous Penning-trap g -factor measurements, in particular, those accomplished at ALPHATRAP setup. This is fine, but I would be mostly interested by the special features of this one. What was the largest problem when going from low to high Z ? Which special techniques allowed the authors to solve it successfully? How far are we from measurements with e.g. lead and uranium?

This is mentioned by both reviewers #1 and #3 and we agree that this is not explained enough in the text.

We added two sentences in the introduction to discuss the main achievement of this measurement, meaning the external production of highly charged ions in the Heidelberg-EBIT which are then loaded and measured in our apparatus (92f).

There is also an added phrase in the conclusion to emphasize this once more (369f).

Measurements with lead or uranium are one of the main goals of our collaboration.

Currently the limiting factor lies in the ion production with the EBIT, which cannot produce much heavier hydrogenlike ions than tin.

Some sentences have been added to the Methods section "Ion Production", which elaborate on this. (445f)

Vacuum problems or spin-flip detection seem to be no issue in the current setup with the cryo valve and the analysis trap.

3. In Figure 4 the level of testing bound-state QED is demonstrated for binding energies (H- and Li-like uranium), HFS (H- and Li-like bismuth) and g -factor - only with H-like systems for some reason. And what about Li-like in this case? Moreover, nothing is said about muonic and pionic atoms - only a couple of dots on the plot. What is the measured quantity in this case?

Great point, the Li-like g factors can surely be added. Fig. 4 has been updated accordingly.

It was not added before as with the additional terms due to the electron-electron interactions, the theoretical access to the g factor is reduced.

Further, the bound-state QED effects are smaller than in the corresponding hydrogenlike system as the $2s$ wavefunction is more spread out from the nucleus.

We also considered to add more measurements as e.g. the Boronlike Argon g -factor measurement, but these many-electron systems have a much lower theoretical access, as the e - e interaction gets even harder to calculate. Further, due to the small Z the QED contribution itself is also small, resulting in a weak resolution for both experiment and theory. Therefore they provide no relevant information for the reader.

A remark to the pionic and muonic measurements is added in the caption, namely that the measured quantity is their Lamb shift in an excited state.

This is also complemented by an additional Extended Data Table which shows the individual values and uncertainties for each data point as suggested by Referee #1.

4. In section 2.6 the calculations of the electron binding energy in Sn within the MCDHF and RCI methods are discussed. First, I suggest giving a hint about how the QED effects are taken into account in this case, i.e., what is the model operator implemented in GRASP. Second, I'm skeptical about the corresponding uncertainty estimation via the ab initio results, since the many-electron effects may be dominant in the case of Sn^{4+} . Luckily, this is not the dominant contribution to the total uncertainty. Third, is the omega-dependence of the Breit interaction taken into account? And finally, a table with all the discussed contributions and uncertainties would be quite useful here.

We thank the Referee for the comments and suggestions. Now, a table with individual contributions to the binding energy of Sn^{4+} , and their uncertainties are added to the corresponding section.

The QED term is accounted for in GRASP via a screened hydrogenic approximation. Thus, the method effectively includes some many-electron QED effects. It gives a value of 79.08 eV for Sn^{4+} , with a 76.64-eV contribution from the $1s^2 2s^2$ electrons of Be-like Sn^{46+} . Considering that this approximation does not treat the many-electron QED effect accurately, we compared our GRASP QED result for Be-like Xe^{50+} to its ab initio value shown in refs. 57 and 58 (ref. 60/61 in the resubmitted manuscript). It is found that our GRASP value is about 0.86% larger than the ab initio value. Therefore, the same relative discrepancy is used to derive the final QED correction of 78.40(68) eV for Sn^{4+} . Nevertheless, all uncalculated QED effects, as well as the contributions from frequency-dependent transverse photon interactions, have already been conservatively included in the systematic errors. To make it clearer to readers, we have now pointed out in the main text that the calculated QED term is based on the screened hydrogenic approximation, and the uncalculated QED effects are effectively included in the systematic errors.

The frequency-dependent transverse photon interactions could not be properly treated by the multiconfiguration RCI method (mainly due to the involvement of correlation orbitals). Thus, in our previous calculations, their effects are effectively included in the systematic errors. To make this clear, we have pointed out in the main text that systematic errors also include effects from frequency-dependent transverse photon interactions. Nevertheless, a single-configuration RCI calculation based on DHF wavefunctions gives a contribution of -1.16 eV from these effects to the binding energy of Sn^{4+} , and has now been discussed in the revised text. Adding this contribution shifts the Sn mass from 117.901 606 975(56)(5) u to 117.901 606 974(56)(5) u, where only the last digit is changed.

5. The g-factor theory part 2.7 is rather chaotic and difficult to comprehend by a non-specialist. It is split into three paragraphs "Nuclear corrections", "Uncertainties of two- and higher-loop QED", and "Muonic and hadronic vacuum polarization". I would expect a step-by-step discussion of the QED effects, from the leading terms to the unknown ones. Instead, only some problematic parts are discussed in an unpredictable order. In general, I suggest to reshape this part, in particular, to comment somehow on all the contributions in Table 3. In the following, some more specific comments are given.

6. The first and largest part "Nuclear corrections" is expected to cover the nuclear size (NS), nuclear recoil (NR) and nuclear polarization (NP) effects. The NS discussion is the most elaborate, and still I can't understand, whether it includes the NS-QED and NS-NR cross terms? Meanwhile, NR is incomplete: nothing is said about the first-order term in the mass ratio. Some aspects of the vacuum polarization are discussed here, while it's not a "nuclear correction", or is it?

7. What was the motivation to pay some special attention to the muonic and hadronic vacuum polarization? Is it strongly enhanced in this case for some reason? Will it be possible to verify these corrections in future improved measurements?

8. Table 3 contains lines with references and those with "adopted from". What is the difference? Which terms were calculated in this work?

Reply to 5-8: Following the referee's recommendation, we are happy to expand our presentation and systematically write about all contributions. The updated text is now arranged into various subsections which systematically cover all g-factor contributions. "Free-electron contributions", "Nuclear effects", "One-loop QED", "Combined QED - Nuclear size effects", "Two- and higher-loop QED", "Muonic and hadronic vacuum polarization".

The section “Nuclear effects” now contains the NS, NR and NP effects. The discussion of the vacuum polarization in the “Nuclear effects” part of our previous version refers to the NS-QED cross terms. In the updated version, the discussion of the NS-QED cross terms was shifted into a subsection of its own.

As for the incomplete discussion of the NR contribution, the following sentence on the first-order in mass ratio term of the NR effect has now been added to the text. “The leading recoil term of first order in the mass ratio was calculated to all orders in $(Z\alpha)$ using formulas and tabulated parameters in Ref. [Shabaev2002].” We also added one sentence about the radiative recoil contribution. So far, recoil contributions to the g -factor are only available for the point-nucleus model, hence the absence of NS-NR cross terms.

We paid special attention to Muonic and Hadronic vacuum polarization contributions in our text because these have not been investigated in detail in the literature. The only available reference on these contributions [Belov2016] does contain several tabulated values for different Z , but not for a Z near 50. It would seem unlikely that muonic or hadronic vacuum polarization can be identified in a g -factor experiment in the near future, considering that the total effect is found to be smaller than e.g. the uncertainty due to the nuclear charge radius. However, especially when considering the possibility for searches of physics beyond the standard model, or the possibility to extract fundamental constants from g -factor measurements, it is important to also understand the small (standard model) g -factor contributions.

The “adopted from” refers to contributions we calculated employing cited methods. In other cases, we either used calculated results from cited references, or we calculated these contributions using formulas from cited references. This distinction is now clearly indicated for each contribution in the updated text. We therefore replaced the “adopted from” label from Table 3 with “this work, method from”, short “TW, methods”.

9. I’m not aware of the corresponding Nature style requirements, but a separate part without any text looks strange to me. Section 2.8 with several tables would benefit strongly from some comments concerning the tables. Otherwise these tables may belong to the corresponding section in which they are referenced.

We agree with the referee that the Extended Data Figures and Tables would be more accessible if they appeared directly at the corresponding section in the Methods. But the Nature Formatting Guide states that the Methods may not contain figures/tables and should be added separately at the end in the Extended Data Section.

Reviewer Reports on the First Revision:

Referees' comments:

Referee #1 (Remarks to the Author):

I am now satisfied with the authors' response. And I notice that the authors also have made a lot of modification according to other referees' comments.